# Symmetry transitions during gating of the TRPV2 ion channel in lipid membranes

Lejla Zubcevic[1], Allen L Hsu[2], Mario J Borgnia[1,2], Seok-Yong Lee[1]*

[1]Department of Biochemistry, Duke University School of Medicine, Durham, United States; [2]Genome Integrity and Structural Biology Laboratory, National Institute of Environmental Health Sciences, National Institutes of Health, Department of Health and Human Services, Research Triangle Park, United States

**Abstract** The Transient Receptor Potential Vanilloid 2 (TRPV2) channel is a member of the temperature-sensing thermoTRPV family. Recent advances in cryo-electronmicroscopy (cryo-EM) and X-ray crystallography have provided many important insights into the gating mechanisms of thermoTRPV channels. Interestingly, crystallographic studies of ligand-dependent TRPV2 gating have shown that the TRPV2 channel adopts two-fold symmetric arrangements during the gating cycle. However, it was unclear if crystal packing forces played a role in stabilizing the two-fold symmetric arrangement of the channel. Here, we employ cryo-EM to elucidate the structure of full-length rabbit TRPV2 in complex with the agonist resiniferatoxin (RTx) in nanodiscs and amphipol. We show that RTx induces two-fold symmetric conformations of TRPV2 in both environments. However, the two-fold symmetry is more pronounced in the native-like lipid environment of the nanodiscs. Our data offers insights into a gating pathway in TRPV2 involving symmetry transitions.
DOI: https://doi.org/10.7554/eLife.45779.001

*For correspondence:
seok-yong.lee@duke.edu

**Competing interests:** The authors declare that no competing interests exist.

## Introduction

Transient Receptor Potential V (TRPV) channels are part of the larger TRP channel family which plays important roles in numerous physiological processes (*Clapham et al., 2001*). A subset of TRPV channels, including subtypes TRPV1-TRPV4, possess an intrinsic capability to sense heat and are therefore referred to as thermoTRPV channels (*Cao et al., 2013a*; *Liu and Qin, 2016*; *Smith et al., 2002*; *Chung et al., 2003*). TRPV1-TRPV4 are non-selective cation channels which play important physiological roles in sensing noxious heat (*Bölcskei et al., 2010*; *Julius, 2013*; *Marwaha et al., 2016*; *Mitchell et al., 2014*), maintaining cardiac structure (*Katanosaka et al., 2014*) and maintaining skin (*Eytan et al., 2014*; *Imura et al., 2009*; *Kim et al., 2016*), hair (*Asakawa et al., 2006*; *Imura et al., 2007*; *Xiao et al., 2008*) and bone physiology (*Masuyama et al., 2008*). A distinctive feature of TRPV1 and TRPV2 is their permeability to large organic cations (*Chung et al., 2008*), such as the cationic dye YO-PRO-1 and the sodium channel blocker QX-314. This feature has led to proposals to utilize these channels as conduits for delivering small molecules to intracellular targets (*Puopolo et al., 2013*). The non-conducting structures of TRPV1 and TRPV2 possess two restrictions, one at the selectivity filter (SF) and second one at the intracellular mouth of the pore (termed the common gate) (*Liao et al., 2013*; *Zubcevic et al., 2016*; *Huynh et al., 2016*). Both restrictions must open widely to accommodate the passage of large organic cations. However, the mechanism that enables such opening was long unclear. In order to study the permeation of both metal ions and large organic cations in TRPV2, we recently crystallized the rabbit resiniferatoxin (RTx)-sensitive (*Zhang et al., 2016*) TRPV2 channel with a truncation in the pore turret in the presence of the agonist RTx (*Zubcevic et al., 2018a*). This study led to the revelation that the binding of RTx leads to a two-fold symmetric (C2) opening at the selectivity filter that is wide enough to permeate YO-PRO-1. This unexpected result offered the first experimental evidence that the homotetrameric TRPV2 can

adopt C2 symmetric conformations during the gating cycle. However, it was unclear if crystal contacts or the crystallization conditions (e.g. high concentration of $Ca^{2+}$) played a role in stabilizing the C2 symmetry. In addition, the minimal TRPV2 construct used in the crystallographic study lacked the pore turret, a region that is not essential for function (*Liao et al., 2013*; *Zubcevic et al., 2016*; *Zubcevic et al., 2018a*; *Yao et al., 2010*) but had previously been shown to have a modulatory effect on gating in TRPV1 and TRPV2 (*Jara-Oseguera et al., 2016*; *Dosey et al., 2019*). It was uncertain if the absence of this region in our crystallographic study affected the symmetry of the channel.

In order to answer these questions and further study the role of two-fold symmetry in TRPV channel gating, we conducted cryo-electronmicroscopy (cryo-EM) studies of the full-length, RTx-sensitive rabbit TRPV2 (*Zhang et al., 2016*) channel reconstituted into nanodiscs and amphipol. We present four structures of the TRPV2/RTx complex, one obtained in nanodiscs (TRPV2$_{RTx-ND}$) and three in amphipol (TRPV2$_{RTx-APOL\ 1-3}$) determined to 3.8 Å, 2.9 Å, 3.3 Å and 4.2 Å resolution, respectively (*Table 1*, *Figure 1*). Our data shows that binding of RTx induces C2 symmetric conformations in TRPV2, but the extent of C2 symmetry depends on the environment in which the channel is reconstituted. C2 symmetry is particularly pronounced in the dataset collected from nanodisc-reconstituted TRPV2, which better approximates the physiological environment of the channel. Moreover, the data offers further insights into the allosteric coupling between the RTx-binding site and the activation gates in TRPV2, confirms the critical role of the S4-S5 linker π-helix (S4-S5$_{\pi-hinge}$) in ligand-

**Table 1.** Data collection and refinement statistics

| Data collection and processing | TRPV2$_{RTx-ND}$ | TRPV2$_{RTx-APOL\ 1}$ | TRPV2$_{RTx-APOL\ 2}$ | TRPV2$_{RTx-APOL\ 3}$ |
|---|---|---|---|---|
| Electron microscope | Titan Krios | | Titan Krios | |
| Electron detector | Falcon III | | Falcon III | |
| Magnification | 75,000x | | 75,000x | |
| Voltage (kV) | 300 | | 300 | |
| Electron exposure (e–/Å$^2$) | 42 | | 42 | |
| Defocus range (μm) | −1.25 to −3.0 | | −1.25 to −3.0 | |
| Pixel size (Å) | 1.08 | | 1.08 | |
| Detector | Counting | | Counting | |
| Total extracted particles (no.) | 1,407,292 | | 580,746 | |
| Refined particles (no.) | 482,602 | | 470,760 | |
| **Reconstruction** | | | | |
| Final particles (no.) | 112,622 | 101,570 | 109,623 | 90,862 |
| Symmetry imposed | C2 | C4 | C2 | C2 |
| Nominal Resolution (Å) | 3.8 | 2.9 | 3.3 | 4.19 |
| FSC 0.143 (masked/unmasked) | 3.7/3.9 | 2.9/3.05 | 3.2/3.5 | 4.0/4.3 |
| Map sharpening *B* factor (Å$^2$) | −30 | −78 | −92 | −133 |
| **Refinement** | | | | |
| **Model composition** | | | | |
| Non-hydrogen atoms | 16,878 | 18,236 | 18,452 | 17,548 |
| Protein residues | 2396 | 2404 | 2440 | 2440 |
| Ligands | RTx: 4 | RTx: 4 | RTx: 4 | RTx: 4 |
| **Validation** | | | | |
| MolProbity score | 1.39 | 1.11 | 1.28 | 1.37 |
| Clashscore | 4 | 1.9 | 2.7 | 2.7 |
| Poor rotamers (%) | 0 | 0 | 0 | 0 |
| **Ramachandran plot** | | | | |
| Favored (%) | 96.5 | 97.1 | 96.6 | 95.5 |
| Allowed (%) | 3.5 | 2.9 | 3.4 | 4.5 |
| Disallowed (%) | 0 | 0 | 0 | 0 |

DOI: https://doi.org/10.7554/eLife.45779.010

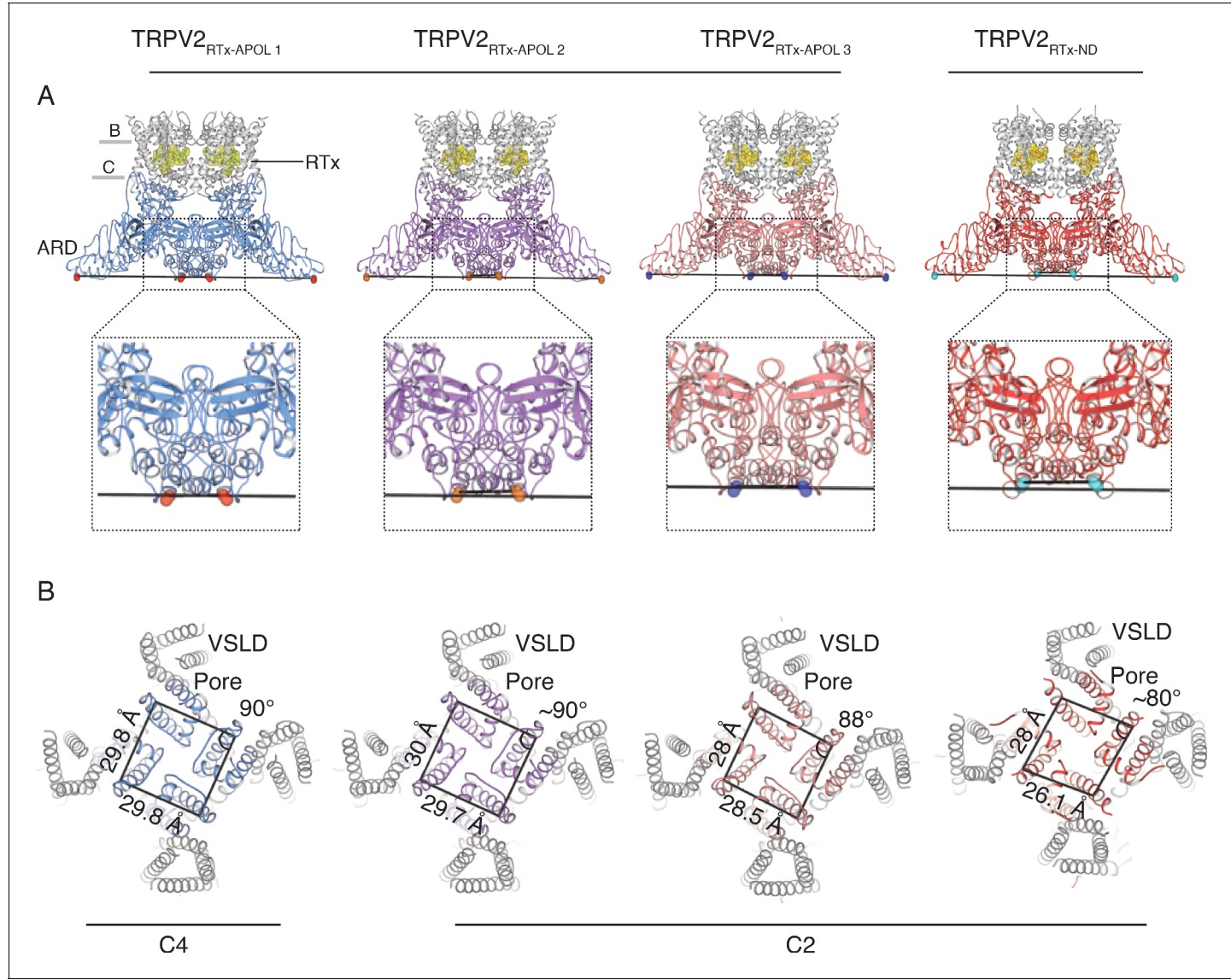

**Figure 1.** Overview of TRPV2$_{RTx-APOL}$ and TRPV2$_{RTx-ND}$ structures. (**A**) Orthogonal view of TRPV2$_{RTx-APOL\ 1-3}$ and TRPV2$_{RTx-ND}$ structures. TM domains are colored in grey and the cytoplasmic domains (ARD and C-terminal domain) are colored in blue, violet, salmon and red, respectively. RTx is shown in stick and sphere representation and colored in yellow. Lines drawn between diagonally opposite ARDs (residue E95, shown in red, orange, blue and cyan spheres, respectively) illustrate the relative position of ARDs in the tetramer. The close-up shows that the ankyrin repeats of diagonally opposing subunits in TRPV2$_{RTx-APOL\ 2}$ and TRPV2$_{RTx-ND}$ are positioned in different planes. (**B**) Top view of the channel (S5, S6 and PH are colored in blue, violet, salmon and red, respectively). Lines drawn between residues V620 in the S6 helix illustrate the symmetry within the pore domain. Distances and angles indicate the presence of two-fold symmetry.

DOI: https://doi.org/10.7554/eLife.45779.002

The following figure supplements are available for figure 1:

**Figure supplement 1.** Cryo-EM data collection and processing, TRPV2$_{RTx-APOL}$.

DOI: https://doi.org/10.7554/eLife.45779.003

**Figure supplement 2.** Cryo-EM data collection and processing, TRPV2$_{RTx-ND}$.

DOI: https://doi.org/10.7554/eLife.45779.004

**Figure supplement 3.** C2 symmetry in the TRPV2$_{RTx-ND}$.

DOI: https://doi.org/10.7554/eLife.45779.005

**Figure supplement 4.** Representative electron densities in the TRPV2$_{RTx-APOL\ 1}$ cryo-EM map.

DOI: https://doi.org/10.7554/eLife.45779.006

**Figure supplement 5.** Representative electron densities in the TRPV2$_{RTx-APOL\ 2}$ cryo-EM map.

DOI: https://doi.org/10.7554/eLife.45779.007

*Figure 1 continued*

**Figure supplement 6.** Representative electron densities in the TRPV2$_{RTx-APOL\ 3}$ cryo-EM map.
DOI: https://doi.org/10.7554/eLife.45779.008
**Figure supplement 7.** Representative electron densities in the TRPV2$_{RTx-ND}$ cryo-EM map.
DOI: https://doi.org/10.7554/eLife.45779.009

dependent gating of TRPV2, and provides a glimpse of the conformational landscape of TRPV2 gating.

## Results

In order to capture the RTx-induced gating transitions in the rabbit TRPV2 channel, we conducted cryo-EM studies of the TRPV2/RTx complex reconstituted into amphipol (TRPV2$_{RTx-APOL}$) and nanodiscs (TRPV2$_{RTx-ND}$). Amphipols (*Zoonens and Popot, 2014*) have been a useful tool in structural studies of membrane proteins, and especially TRP channels (*Liao et al., 2013*; *Zubcevic et al., 2016*; *Cao et al., 2013b*; *Paulsen et al., 2015*; *Yoo et al., 2018*; *Hirschi et al., 2017*; *Zubcevic et al., 2018b*). Indeed, Amphipol A8-35 enabled the very first structural determination of the TRPV2 channel (*Zubcevic et al., 2016*). Nanodiscs, on the other hand, represent the closest in vitro approximation to the native lipid membranes used in structural studies (*Denisov and Sligar, 2016*). The data was processed using RELION (*Scheres, 2012*) (Materials and methods), with no symmetry imposed during particle classification and 3D reconstruction in order to avoid obscuring any classes with lower symmetry (C1 and C2) that might exist in the sample. Symmetry was only imposed in the last step of the refinement and only if the 3D reconstructions showed clear two-fold (C2) or four-fold (C4) symmetry (*Figure 1—figure supplements 1–3*). Classification of the TRPV2$_{RTx-APOL}$ sample revealed the presence of four classes: one low-resolution (~7 Å) class, which was excluded from further analysis, and three higher resolution classes which are representative of three different conformations. These include one C4 symmetric and two distinct C2 symmetric classes refined to 2.9 Å, 3.3 Å and 4.2 Å, respectively (*Figure 1*, *Figure 1—figure supplement 1*). By contrast, 3D classification of the TRPV2$_{RTx-ND}$ converged on a single C2 symmetric conformation resolved to 3.8 Å (*Figure 1*, *Figure 1—figure supplements 2–3*). All four maps were of sufficient quality to enable placement of individual structural motifs with confidence (*Figure 1—figure supplements 4–7*) and the models for all four structures were built to good overall geometry (*Table 1*).

### The transmembrane domains of TRPV2$_{RTx-APOL}$ are trapped in a closed conformation

Unexpectedly, the transmembrane domains (TM) of the three structures obtained from amphipol-reconstituted TRPV2, TRPV2$_{RTx-APOL\ 1-3}$, show similarity to our previously solved cryo-EM structure of TRPV2 in its apo form (*Zubcevic et al., 2016*) (TRPV2$_{APO}$) and adopt non-conducting conformations (*Figure 2—figure supplement 1*). While fully bound to RTx, the TM domains of TRPV2$_{RTx-APOL\ 1}$ and TRPV2$_{RTx-APOL\ 2}$ structures largely retain C4 symmetry (*Figure 1* and *Figure 3—figure supplement 1*). However, the TMs of TRPV2$_{RTx-APOL\ 3}$ exhibit a slight departure from C4 symmetry in the pore (*Figure 3—figure supplement 2*). The effects of RTx on the TRPV2$_{RTx-APOL}$ are particularly obvious in the ankyrin repeat domains (ARD) of the two-fold symmetric TRPV2$_{RTx-APOL\ 2}$ and TRPV2$_{RTx-APOL\ 3}$ which display pronounced broken symmetry and a range of rotational states (*Figure 1A*, *Figure 3—figure supplements 1–3*).

In order to determine the effect of RTx on the TRPV2$_{RTx-APOL}$ sample, we aligned TRPV2$_{RTx-APOL\ 1}$ with TRPV2$_{APO}$. The transmembrane helices S1-S6 of the two channels aligned remarkably well (Cα R.M.S.D = 0.86) (*Figure 3—figure supplement 1*). However, RTx binding induces a 5° clockwise rotation of the ARD when viewed from the extracellular space and a ~ 10 Å lateral widening of the cytoplasmic assembly (*Figure 3—figure supplement 1*). In addition, RTx causes a conformational change in the S4-S5 linker (*Figure 3—figure supplement 1*), as well as a displacement of the TRP domain (*Figure 3—figure supplement 1*). The conformational change in the S4-S5 linker is caused by the introduction of a π-helical turn at the junction of the S4-S5 linker and the S5 helix in the TRPV2$_{RTx-APOL\ 1}$ structure (S4-S5$_{π-hinge}$), which is absent in TRPV2$_{APO}$ (*Figure 3—figure supplements 1 and 4*). This observation concurs with our previous finding that RTx binding elicits a conformational

change in the S4-S5 linker, and that the S4-S5$_{\pi\text{-hinge}}$ is critical for ligand-dependent gating in TRPV2 (*Zubcevic et al., 2018a*). In TRPV2$_{\text{RTx-APOL 3}}$, slight C2 symmetry is observed in the TM domains and is evident in the SF, PH and the S4-S5 linker (*Figure 3—figure supplement 2*). Nevertheless, the RTx-induced conformational changes in the S4-S5 linker are not efficiently propagated to the TM in the TRPV2$_{\text{RTx-APOL}}$ structures, and they fail to open either of the two restrictions in the pore (*Figure 2—figure supplement 1*). Instead, RTx only effects changes in its immediate binding site above the S4-S5 linker and in the parts of the channel not bound by amphipol, strongly suggesting that the polymer constricts the TM and prevents conformational changes at the S4-S5 linker and the ARD from propagating to the TM domain. The fact that the TRPV2/RTx complex is stabilized in multiple distinct closed states with different arrangements of the ARD assembly (*Figure 1*, *Figure 3—figure supplements 1–3*) suggests that the conformational changes in the ARD might represent low-energy, pre-open states that can be achieved without substantial changes in the TM domains.

Interestingly, metal ions are not visualized in the pores of any of the TRPV2$_{\text{RTx-APOL}}$ structures, despite the high resolutions obtained in this study. Whether this is the result of cryo-EM experimental conditions is unclear, but thus far metal ions occupying the SF and the pores of thermoTRPV channels have only been captured in structures obtained by X-ray crystallography (*Zubcevic et al., 2018a*).

## RTx induces a break in symmetry in TRPV2$_{\text{RTx-ND}}$

In stark contrast to the amphipol-reconstituted channel, RTx binding induces C2 symmetry in the nanodisc-reconstituted TRPV2 which extends throughout the channel. The symmetry of the TRPV2$_{\text{RTx-ND}}$ map was assessed both visually, and by the *Map Symmetry* function in Phenix which yielded a CC = 0.9 and score of 1.28 for C2 symmetry. For comparison, C4 symmetry yielded a lower correlation coefficient (CC = 0.8). To further confirm the correctness of the symmetry assignment, we evaluated the fit of the TRPV2$_{\text{RTx-ND}}$ model built into the C2 symmetric map to the non-symmetrized C1 map (*Figure 1—figure supplement 3*). In addition, we evaluated the fit of the TRPV2$_{\text{RTx-ND}}$ model to the individually refined non-symmetrized classes 1 and 6, which constitute the TRPV2$_{\text{RTx-ND}}$ map (*Figure 1—figure supplement 3*). All FSC curves indicate that the two-fold symmetric model fits well into the C1 maps and the density of the C1 symmetric TRPV2$_{\text{RTx-ND}}$ map supports the model (*Figure 1—figure supplement 3*), showing that two-fold symmetry is truly present in the TRPV2$_{\text{RTx-ND}}$ sample.

The pore of TRPV2$_{\text{RTx-ND}}$ adopts a C2 symmetric arrangement (*Figure 2A*). The pore helices are arranged so that the carbonyl oxygens of the selectivity filter in subunits B and D line the entry to the pore while pore helices of subunits A and C tilt away from the permeation pathway. This arrangement creates a large C2 symmetric opening where the narrowest constriction between SF residues in diametrically opposing subunits A and C and B and D is ~11 Å and ~8 Å, respectively. This results in an SF with ample room to accommodate large organic cations (*Figure 2B*). A closer look at the pore helices reveals that this arrangement in the SF is achieved through a ~ 10° swivel of the subunit A pore helix, which brings the N-terminal part of the helix closer to S5 while distancing it from S6 (*Figure 2C*). The position of the pore helices controls the size and the shape of the SF, and appears to exert control over ion permeation in TRPV2. While the SF is widely open, the common gate adopts a putative intermediate conformation where two of the diagonally opposing subunits adopt a closed state, and the remaining two are open. In subunits A and C, the S6 helix adopts a straight α-helical, closed conformation, while the S6 of subunits B and D is bent and the common gate apparently open (*Figure 2A*). However, the overall functional state of the common gate is likely non-conductive as the gate residues from subunits A and C would presumably hinder ion permeation.

In order to establish the origin of the C2 symmetry in the TRPV2$_{\text{RTx-ND}}$ structure, we aligned subunits A and B (Cα R.M.S.D = 0.96) (*Figure 3—figure supplement 5*). Similar to our previous findings, this alignment shows that the two subunits diverge at the S4-S5 linker and the PH and indicates that rotation of subunits around the S4-S5$_{\pi\text{-hinge}}$ appears to result in the distinct C2 symmetric arrangement observed in TRPV2$_{\text{RTx-ND}}$ (*Figure 3—figure supplements 4–5*).

When compared to the TRPV2$_{\text{APO}}$, the TM domains of the TRPV2$_{\text{RTx-ND}}$ structure appear to contract in an asymmetric manner (*Figure 3A*), while the ARD assembly expands by ~10 Å and rotates by 3° (*Figure 3B*). The TM domains and the ARDs appear to move as a single rigid body, which is evident when individual subunits from TRPV2$_{\text{APO}}$ and TRPV2$_{\text{RTx-ND}}$ are superposed (Cα R.M.S.

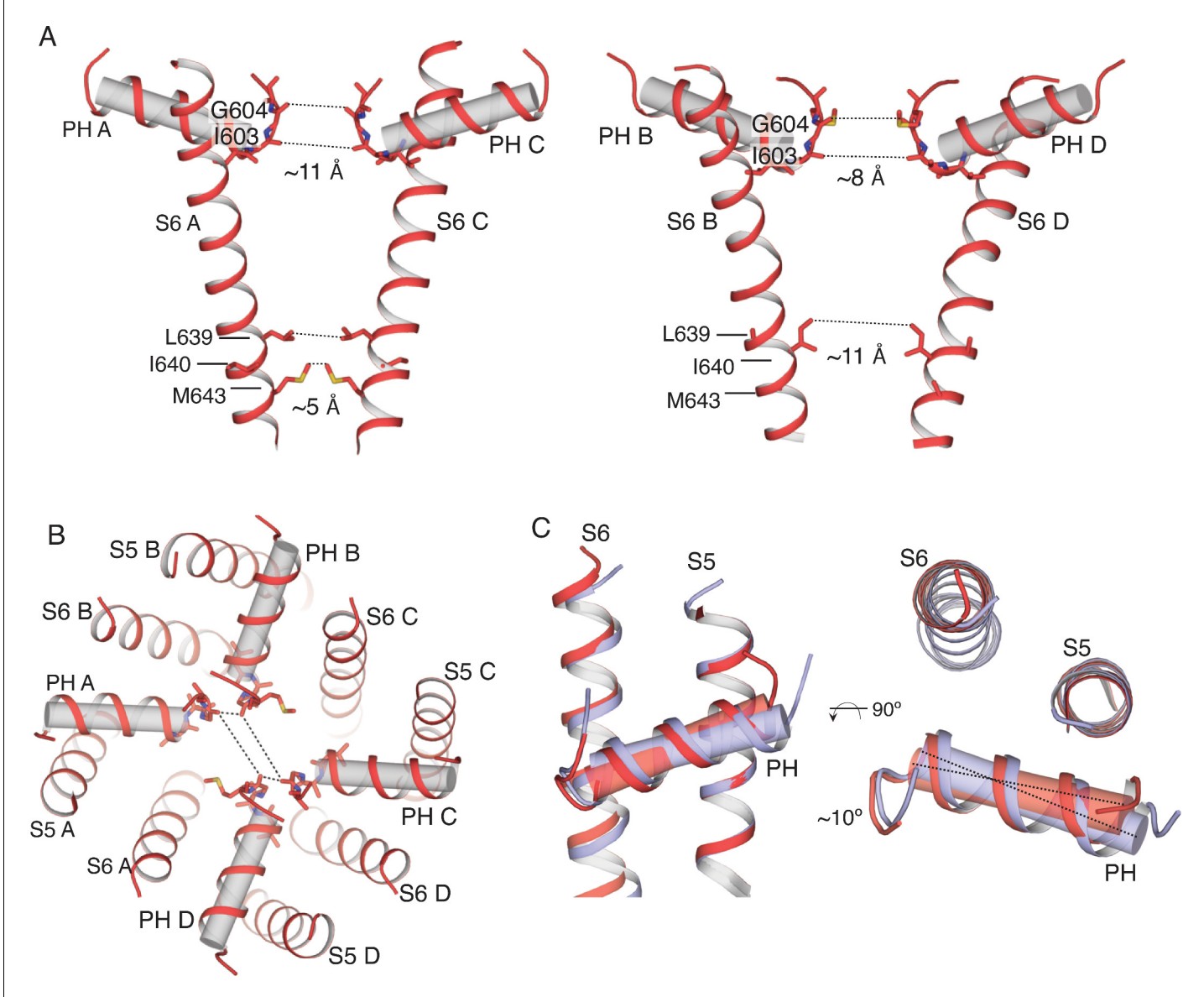

**Figure 2.** Overview of the pore in the TRPV2$_{RTx-ND}$ structure. (**A**) S6 and pore helices of subunits A and C (left) and subunits B and D (right). Pore helices are shown in both cartoon and cylinder representation (grey). Dashed lines and values represent distances between the indicated residues. S6 helices in A and C are straight and α-helical, while the S6 in subunits B and D is bent. (**B**) Top view of the TRPV2$_{RTx-ND}$ pore, with pore helices shown in both cartoon and cylinder representation. Dashed lines illustrate the distances between residues G604 in the selectivity filter. (**C**) Overlay of the TRPV2$_{RTx-ND}$ pore domains (S5, S6 and pore helices). Subunit A is shown in red and subunit B in violet. The pore helix of subunit A swivels by ~10° relative to subunit B.

DOI: https://doi.org/10.7554/eLife.45779.011

The following figure supplement is available for figure 2:

**Figure supplement 1.** Pore comparison of TRPV2$_{APO}$ (orange) and TRPV2$_{RTx-APOL\ 1-3}$ (blue, purple and salmon, respectively).

DOI: https://doi.org/10.7554/eLife.45779.012

D = 1.9 Å) to reveal that only the S4-S5 linker and the pore helix deviate significantly in the two structures (*Figure 3C*). This coupled movement of the TM and ARD indicates that RTx-binding to TRPV2 in lipid membranes induces a rigid-body rotation of the entire subunit that originates at the S4-S5$_{\pi\text{-hinge}}$ (*Figure 3D–E*).

Interestingly, the TRPV2$_{RTx-ND}$ structure exhibits different degrees of reduced symmetry from the previously determined crystal structure of TRPV2 in complex with RTx (TRPV2$_{RTx-XTAL}$)

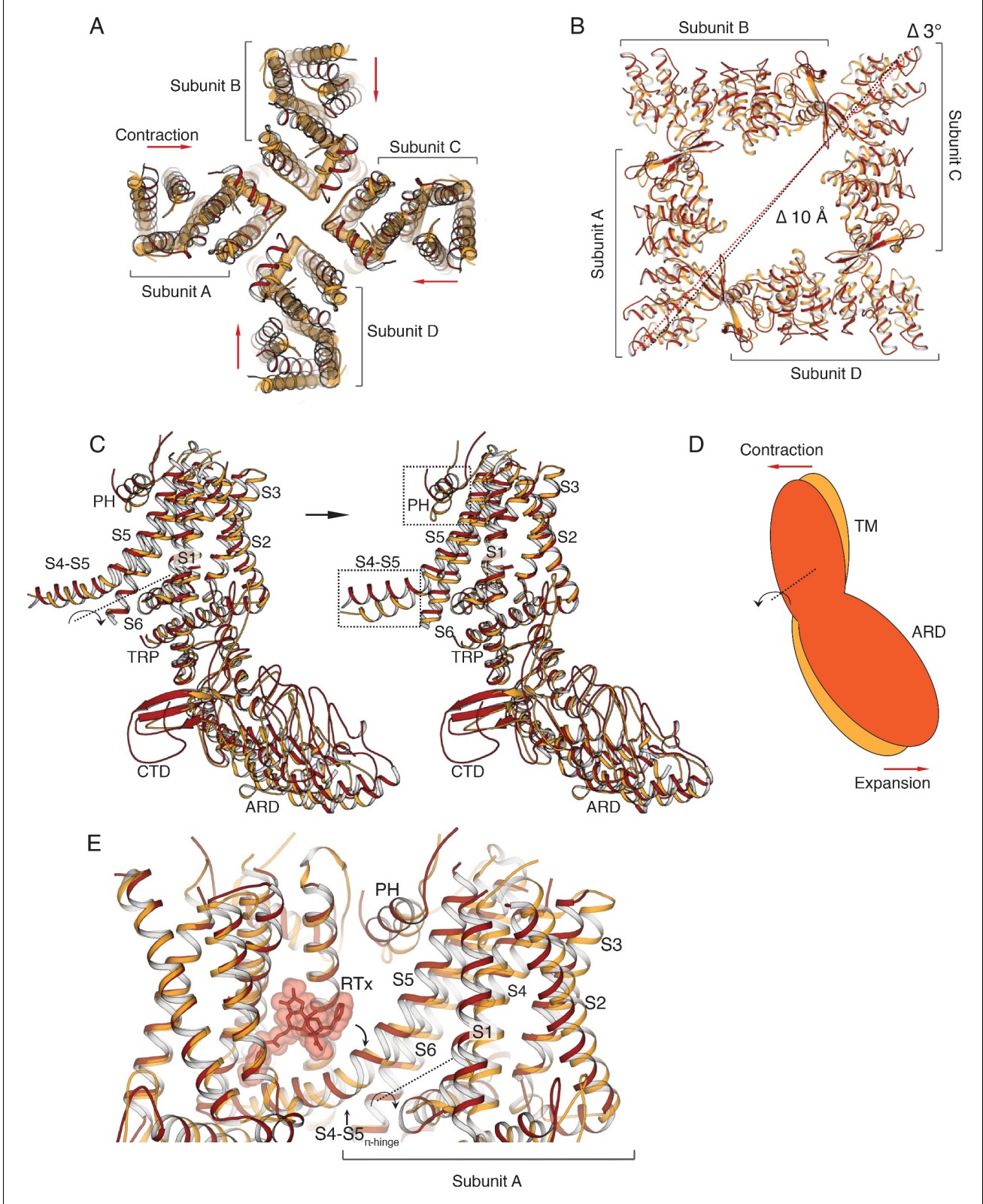

**Figure 3.** Comparison of TRPV2$_{RTx-ND}$ (red) and TRPV2$_{APO}$ (orange). (**A**) Overlay of TRPV2$_{RTx-ND}$ and TRPV2$_{APO}$, top view. TRPV2$_{RTx-ND}$ is shown in cartoon representation and TRPV2$_{APO}$ as cylinders. Relative to TRPV2$_{APO}$, the TM subunits of TRPV2$_{RTx-ND}$ exhibit contraction (red arrows). (**B**) Top view of the ARDs in TRPV2$_{RTx-ND}$ and TRPV2$_{APO}$. TM helices are removed for ease of viewing. Dashed lines represent distances between residues T100, showing a 10 Å expansion ($\Delta$ 10 Å) and 3° rotation ($\Delta$ 3°) of the TRPV2$_{RTx-ND}$ ARD assembly relative to TRPV2$_{APO}$. (**C**) A rigid-body rotation of TRPV2$_{RTx-}$

*Figure 3 continued on next page*

*Figure 3 continued*

$_{ND}$ subunit B around the S4-S5 linker achieves alignment with the subunit B from TRPV2$_{APO}$. Following alignment, only the S4-S5 linkers and the pore helices (PH) diverge in the two subunits (dashed box). (**D**) Cartoon illustrating how the movements of the TM and the ARD in TRPV2$_{RTx-ND}$ are coupled. The red and orange shapes represent a single subunit of TRPV2$_{RTx-ND}$ and TRPV2$_{APO}$, respectively. The rotation of the subunit is manifested as 'contraction' in the TM domains and 'expansion' of the ARD. (**E**) RTx binding in the vanilloid binding pocket exerts force on the S4-S5 linker, changing the conformation of the junction from α- to π-helix, and induces the rotation of the subunit around the S4-S5$_{\pi\text{-hinge}}$.

DOI: https://doi.org/10.7554/eLife.45779.013

The following figure supplements are available for figure 3:

**Figure supplement 1.** Comparison of TRPV2$_{RTx-APOL\ 1-2}$ and TRPV2$_{APO}$.
DOI: https://doi.org/10.7554/eLife.45779.014
**Figure supplement 2.** Two-fold symmetry in TRPV2$_{RTx-APOL\ 3}$ (salmon).
DOI: https://doi.org/10.7554/eLife.45779.015
**Figure supplement 3.** Symmetry breaking in the TRPV2$_{RTx-APOL\ 2-3}$ ARD.
DOI: https://doi.org/10.7554/eLife.45779.016
**Figure supplement 4.** Electron density around the S4-S5 linker π-helices in TRPV2$_{RTx-APOL\ 1}$ (**A**), TRPV2$_{RTx-APOL\ 2}$ (**B**) and TRPV2$_{RTx-APOL\ 3}$ (**C**).
DOI: https://doi.org/10.7554/eLife.45779.017
**Figure supplement 5.** Comparison of TRPV2$_{RTx-ND}$ subunits A (red) and B (violet).
DOI: https://doi.org/10.7554/eLife.45779.018

(*Zubcevic et al., 2018a*). Compared to the TRPV2$_{RTx-XTAL}$, the TM domains of TRPV2$_{RTx-ND}$ contract in an two-fold symmetric manner (*Figure 4A*). This conformational change, which stems from rotation of individual TRPV2$_{RTx-ND}$ subunits around the S4-S5$_{\pi\text{-hinge}}$ (*Figure 4—figure supplement 1*), results in an overall fold that is closer to C4 symmetry than that of the TRPV2$_{RTx-XTAL}$ (*Figure 4B*). However, while the TRPV2$_{RTx-ND}$ helices S1-S6 adopt a more C4 symmetric arrangement, the pore helices and the SF remain distinctly C2 symmetric (*Figure 4C*). Remarkably, the SF of TRPV2$_{RTx-ND}$ is wider than that of TRPV2$_{RTx-XTAL}$, and the two structures display different C2 symmetric openings at the SF (*Figure 4C*). The two different conformations result from both the different arrangements of subunits and changes in the position and tilt angle of the pore helices (*Figure 4D–E*). In the TRPV2$_{RTx-XTAL}$ structure, the pore helices of subunits B and D, which assume a widened conformation, are free of interactions with the pore domain, while a network of interactions (presumably hydrogen bonds) between Y542-T602-Y627 in subunits A and C tethers the pore helices to S5 and S6. Our previous work showed that disruption of these interactions is detrimental to the permeation of large organic cations but has no effect on permeation of metal ions (*Zubcevic et al., 2018a*). Interestingly, the putative hydrogen bond triad is disrupted in all four subunits of the TRPV2$_{RTx-ND}$ structure (*Figure 4—figure supplement 2*). Nevertheless, the SF assumes a fully open state that can potentially accommodate passage of a large cation. This suggests that the putative hydrogen bond triad, while not a feature of the fully open SF, is an essential part of the transition between closed and open states of the channel.

It is interesting to point out that extensive rearrangements around the SF and the PH during gating have thus far only been observed in structural studies of TRPV1 (*Cao et al., 2013b*) and TRPV2 (*Zubcevic et al., 2018a*) channels. In the non-conductive state, the SFs of the remaining members of the TRPV subfamily (TRPV3-TRPV6 [*Zubcevic et al., 2018b*; *Singh et al., 2018*; *Deng et al., 2018*; *Hughes et al., 2018*; *McGoldrick et al., 2018*]) adopt a conformation that is wide enough to accommodate a semi-hydrated cation, and do not move appreciably during channel activation. This may indicate that TRPV1 and TRPV2 are the only members of the TRPV subfamily that possess a gate at the SF, and that the coupling of structural elements necessary for activation of these channels differs from that of TRPV3-TRPV6 (*Zhang et al., 2019*).

As observed in our previous study (*Zubcevic et al., 2018a*), RTx assumes different binding poses in subunits of the C2-symmetric structures, both in the amphipol and the nanodisc samples (*Figure 4—figure supplement 3*) which may lead to the distinct conformations observed in these channels.

Despite the use of a full-length rabbit TRPV2 construct in this study, we were not able to confidently resolve the entire loop connecting S5 to the pore helix known as the 'pore turret'. Interestingly, a recent structure of rat TRPV2 with the pore turret resolved showed that this region, which contains a large number of charged and polar residues, occupies the space within the membrane

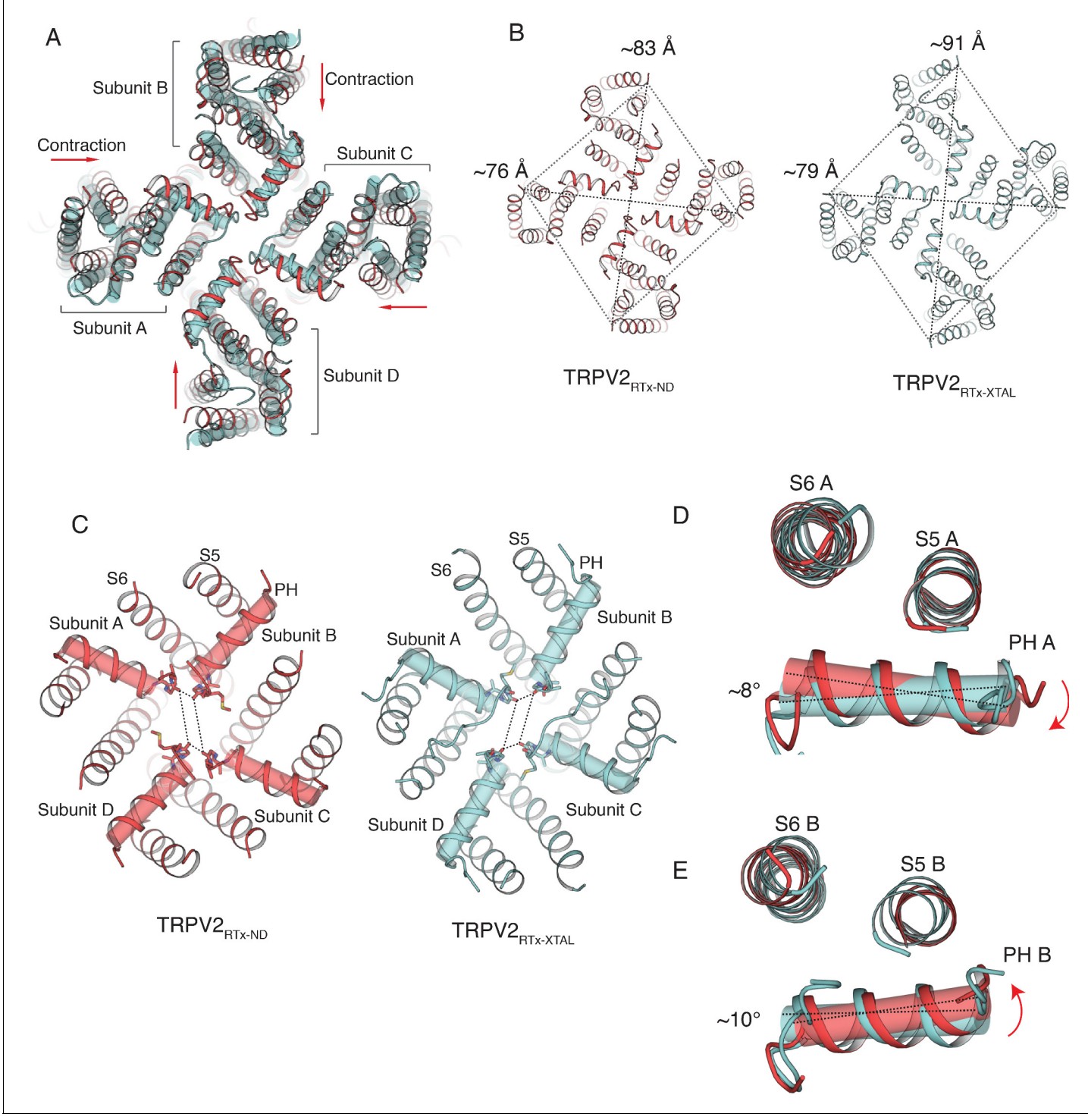

**Figure 4.** Comparison of TRPV2$_{RTx-ND}$ (red) and TRPV2$_{RTx-XTAL}$ (cyan). (**A**) Overlay of TRPV2$_{RTx-ND}$ and TRPV2$_{RTx-XTAL}$, top view. TRPV2$_{RTx-ND}$ is shown in cartoon representation and TRPV2$_{RTx-XTAL}$ as cylinders. Relative to TRPV2$_{RTx-XTAL}$, the TM domains of TRPV2$_{RTx-ND}$ are contracted (red arrows). (**B**) Comparison of two-fold symmetry in TRPV2$_{RTx-ND}$ and TRPV2$_{RTx-XTAL}$. Dashed lines represent distances between residues A427. The distances between diagonally opposing subunits are indicated. (**C**) Top view of the SF in TRPV2$_{RTx-ND}$ and TRPV2$_{RTx-XTAL}$. Pore helices are shown in both cartoon and cylinder representation. Dashed lines represent distances between residues G604 in the selectivity filter. (**D–E**) Overlay of the pore domains of TRPV2$_{RTx-ND}$ and TRPV2$_{RTx-XTAL}$ subunit A (**D**) and subunit B (**E**) shows that the pore helices A and B in TRPV2$_{RTx-ND}$ swivel by ~8° and ~10°, respectively, compared to TRPV2$_{RTx-XTAL}$.

DOI: https://doi.org/10.7554/eLife.45779.019

*Figure 4 continued on next page*

*Figure 4 continued*

The following figure supplements are available for figure 4:

**Figure supplement 1.** Comparison of subunits B in TRPV2$_{RTx-ND}$ (red) and TRPV2$_{RTx-XTAL}$ (cyan).

DOI: https://doi.org/10.7554/eLife.45779.020

**Figure supplement 2.** Interactions between the pore helix (PH) and S5 and S6.

DOI: https://doi.org/10.7554/eLife.45779.021

**Figure supplement 3.** Binding of RTx in TRPV2$_{RTx-APOL}$ and TRPV2$_{RTx-ND}$.

DOI: https://doi.org/10.7554/eLife.45779.022

**Figure supplement 4.** The pore turret in TRPV2$_{RTx-APOL\ 1}$ (blue) and rat TRPV2 (PDB 6BO4, purple).

DOI: https://doi.org/10.7554/eLife.45779.023

**Figure supplement 5.** The common gate in TRPV2$_{RTx-ND}$ (red) and TRPV2$_{RTx-XTAL}$ (cyan).

DOI: https://doi.org/10.7554/eLife.45779.024

**Figure supplement 6.** The amphipol and nanodisc clouds.

DOI: https://doi.org/10.7554/eLife.45779.025

plane between S5 and the Voltage Sensor-Like Domain (VSLD) (*Dosey et al., 2019*). While the density in our cryo-EM maps was not of sufficient quality to build the entire pore turret with confidence, we do observe density following the S5 helix and preceding the pore helix. However, the direction of this density is perpendicular to the membrane and does not agree with the structure reported for rat TRPV2 (*Figure 4—figure supplement 4*). Indeed, the pore turret is the least conserved region amongst the TRPV2 orthologs, and the variations in its sequence might indicate that the turret adopts different conformations in TRPV2 channels from different species. Nevertheless, our study clearly shows that the omission of this region from the construct used in the crystallographic study of the TRPV2/RTx complex is not the cause of the C2 symmetry.

While both TRPV2$_{RTx-ND}$ and TRPV2$_{RTx-XTAL}$ structures adopt C2 symmetry, the distinct arrangement of subunits within the two channels suggests that the structures represent different functional states. We propose that TRPV2$_{RTx-XTAL}$ precedes TRPV2$_{RTx-ND}$ in the conformational activation trajectory based on two observations. Firstly, the common gate is fully closed in the TRPV2$_{RTx-XTAL}$ while it adopts an apparently partially open conformation in TRPV2$_{RTx-ND}$ (*Figure 4—figure supplement 5*). Secondly, our previous studies have shown that the putative hydrogen bond network between S5 and S6 and the pore helix is essential for the channel's ability to transition to a fully open SF that can accommodate large organic cations (*Zubcevic et al., 2018a*). Nevertheless, in TRPV2$_{RTx-ND}$ the pore helices do not interact with S5 and S6 and the SF is fully open. We therefore propose that the conformational step that requires the presence of the putative hydrogen bond triad precedes the open SF conformation seen in TRPV2$_{RTx-ND}$.

## Discussion

Here, we have conducted a study that reveals symmetry transitions associated with gating of the TRPV2 channel by RTx. Interestingly, our data shows that RTx induces C2 symmetric conformations of TRPV2 in both amphipol and nanodiscs, and it thereby negates the hypothetical role of crystallization artefacts and crystal packing bias in stabilising two-fold symmetry. Similarly, C2 symmetry in TRPV2 is independent of the presence or absence of the pore turret region, suggesting that this region does not play an essential role in the regulation of the SF in rabbit TRPV2. Our study, similar to a previously published study of the magnesium channel CorA (*Matthies et al., 2016*), also emphasizes the notion that careful inspection of the intermediate maps and conservative application of symmetry during refinement of cryo-EM data can result in valuable insights into gating transitions and intermediate states. In addition, we have also investigated how amphipols and nanodiscs affect the conformational space that can be accessed during ligand gating of TRPV2.

While both TRPV2$_{RTx-APOL}$ and TRPV2$_{RTx-ND}$ are C2 symmetric, the two-fold symmetry in TRPV2$_{RTx-APOL}$ is confined to regions that are not bound by the amphipol polymer. This is evident in the fact that the TM domains, which are in contact with the amphipol, largely retain four-fold symmetry and the common gate and the SF remain firmly closed, while the ARD exhibit symmetry breaking, rotation and lateral expansion. These data, while adding valuable data points to the conformational landscape of TRPV2, also illustrate the potential caveats of using amphipols in

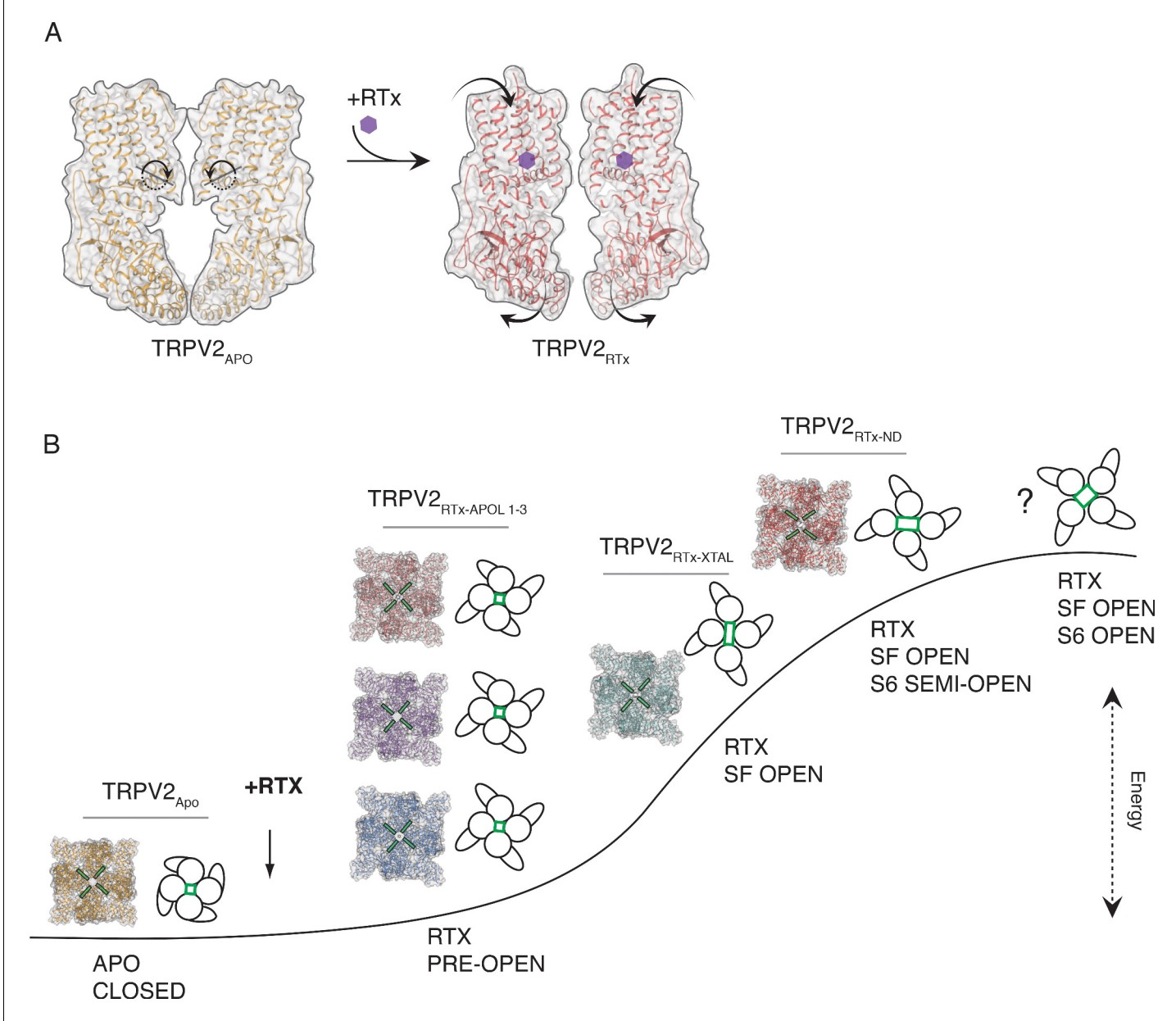

**Figure 5.** Conformational states associated with RTx-mediated gating of TRPV2. (**A**) TRPV2 subunit rotation upon binding of RTx. Rotation axis and direction are indicated in dashed line and circular arrow in apo TRPV (left). The rotation results in contraction of the TM domains and widening of the cytoplasmic assembly (right). (**B**) Hypothetical trajectory of TRPV2 gating with associated conformational states. Upon addition of RTx, TRPV2 first enters low-energy pre-open states that are characterized by rotation, widening and symmetry breaking in the ARD (TRPV2$_{RTx-APOL\ 1-3}$, models shown in cartoon and surface representation). In the next step, the channel assumes C2 symmetric state with an open SF, but closed common (S6) gate (TRPV2$_{RTx-XTAL}$, model shown in cartoon and surface representation). This is followed by a less C2 symmetric state with an open SF and semi-open common (S6) gate (TRPV2$_{RTx-ND}$, model shown in cartoon and surface representation). Finally, we propose that the channel assumes a high-energy fully open state that is C4 symmetric but might have C2 symmetry in the SF. The SF is indicated in green in models and cartoons.

DOI: https://doi.org/10.7554/eLife.45779.026

studies of conformational changes in the transmembrane domains of proteins, as they appear to constrict the TM domains and stabilize low-energy pre-open states (***Figure 4—figure supplement***

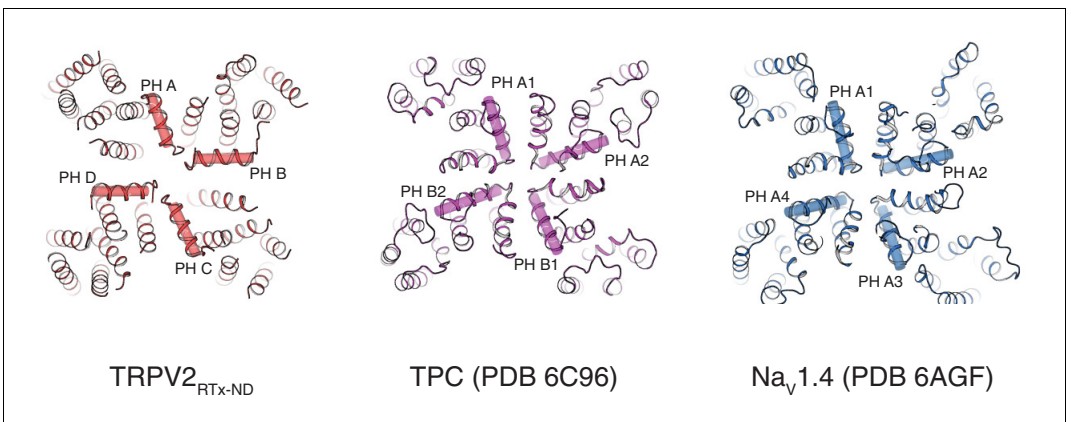

**Figure 6.** Comparison of TRPV2$_{RTx-ND}$ (red), TPC1 (PDB 6C96, purple) and Na$_V$1.4 (PDB 6AGF, blue). Top view, pore helices are indicated.

DOI: https://doi.org/10.7554/eLife.45779.027

*6*). However, at this time we caution against any generalized conclusions about the effect of amphi-pols and look forward to more systematic studies that will address these issues in the future. The TRPV2$_{RTx-ND}$ dataset yielded a single, two-fold symmetric structure thus giving strong evidence that RTx stabilizes two-fold symmetric conformational states in the TRPV2 channel in lipid membranes. The ARDs in the TRPV2$_{RTx-ND}$ structure echo the conformational changes observed in TRPV2$_{RTx-APOL}$. However, in nanodiscs TRPV2 is captured with its SF fully open and its common gate in an intermediate conformation where the gate is apparently open in one set of diagonally opposing subunits and closed in the other. In this structure, the opening of the SF occurs according to a mechanism previously observed in the crystallographic study of the TRPV2/RTx complex where RTx binding in the vanilloid pocket, above the S4-S5$_{\pi-hinge}$, induces a rigid body rotation of the entire subunit. In turn, the rotation causes a break in the interaction network between the pore helix and helices S5 and S6, allowing the pore helices to reposition and the SF to open (*Zubcevic et al., 2018a*).

Interestingly, however, the TRPV2$_{RTx-ND}$ structure differs from the previously obtained TRPV2$_{RTx-XTAL}$. While both structures assume C2 symmetric conformations, the TRPV2$_{RTx-ND}$ channel appears to make a return toward C4 symmetry. Because the SF in TRPV2$_{RTx-ND}$ is fully open, and the common gate adopts an apparently partially open conformation, we reason that TRPV2$_{RTx-ND}$ follows the TRPV2$_{RTx-XTAL}$ structure in the conformational trajectory of the channel. Therefore, it is possible that TRPV2, as it travels toward the final open state where both the SF and the common gate are fully open, would adopt further conformations that increasingly approximate C4 symmetry (*Figure 5*). However, it is interesting to note that while the overall fold of TRPV2$_{RTx-ND}$ indeed is more C4 symmetric than that of TRPV2$_{RTx-XTAL}$, the extent of C2 symmetry is not diminished in its SF. Because the symmetry of the SF does not appear to be dictated by the symmetry of the overall channel, we cannot exclude the possibility that the final open state might indeed possess a C2 symmetric SF while otherwise adopting a nearly C4 symmetric conformation. Our previous functional studies have suggested that C2 symmetric states play a role in the permeation of large organic cations and consequently in the full opening of the SF (*Zubcevic et al., 2018a*). Hence, the channel might be utilizing C2 symmetric states as means to achieve full opening in a step-wise manner. Similar C2 symmetric states elicited by ligand binding have been observed in TRPV3 (*Zubcevic et al., 2018b*) and TRPM2 (*Yin et al., 2018*) channels, which opens up the possibility that C2 symmetry might be widely associated with gating in members of the TRP channel superfamily. Intriguingly, a recent cryo-EM study of the human BK channel reconstituted in liposomes showed that this channel also enters C2 symmetric states (*Tonggu and Wang, 2018*), suggesting that two-fold symmetry might also play a role in the molecular mechanisms of other tetrameric ion channels.

Two-fold symmetry is a well-stablished feature of mammalian Na$^+$ selective Two Pore Channels (TPCs) and Voltage Gated Sodium channels (Na$_V$) (*She et al., 2018*; *Shen et al., 2017*; *Pan et al., 2018*; *Shen et al., 2018*). Interestingly, the arrangement of pore helices in TRPV2$_{RTx-ND}$ resembles that observed in TPC and Na$_V$ (*Figure 6*) and the selectivity filters in all three channels form a 'coin-

slot' (*Hille, 1971*) opening. However, while the selectivity filters of TPC and Na$_V$ remain static during channel gating in order to maintain the structure necessary for Na$^+$ selectivity, the SF of TRPV2 displays a large degree of plasticity. Moreover, the two-fold symmetry observed in TRPV2 is unique in that it arises in response to conformational changes in the TM domains induced by ligand binding. By contrast, the two-fold symmetry in TPC and Na$_V$ stems from the arrangement of their respective homologous tandem repeats.

# Materials and methods

## Key resources table

| Reagent type (species) or resource | Designation | Source or reference | Identifiers | Additional information |
|---|---|---|---|---|
| Cell line | DH10Bac *E. coli* | ThermoFisher Scientific | 10361012 | |
| Cell line | Sf9 | ATCC | CRL-1711 | RRID:CVCL_0549 |
| Recombinant DNA reagent | rabbit TRPV2 | Genscript | Pubmed Accession No. XM_017349044 | |
| Recombinant DNA reagent | Bac-to-Bac Baculovirus Expression System | ThermoFisher Scientific | 10359016 | |
| Recombinant DNA reagent | MSP2N2 scaffold protein | Stephen Sligar laboratory | Addgene:Cat#29520 | PMID:20817758 |
| Chemical compound, drug | *n*-dodecyl-β-d-maltopyranoside(DDM) | Anatrace | D310 | |
| Chemical compound, drug | Cholesteryl Hemisuccinate | Anatrace | CH210 | |
| Chemical compound, drug | Amphipol A8-35 | Anatrace | A835 | |
| Chemical compound, drug | TRIS | Fisher Scientific | BP152 | |
| Chemical compound, drug | NaCl | Fisher Scientific | S271 | |
| Chemical compound, drug | CaCl2 | Fisher Scientific | C70 | |
| Chemical compound, drug | leupeptin | GoldBio | L-010 | |
| Chemical compound, drug | pepstatin | GoldBio | P-020 | |
| Chemical compound, drug | aprotinin | GoldBio | A-655 | |
| Chemical compound, drug | DNase I | GoldBio | D-301 | |
| Chemical compound, drug | β-mercapto ethanol | Sigma Aldrich | M3148 | |
| Chemical compound, drug | PMSF | Sigma Aldrich | P7626 | |
| Chemical compound, drug | anti-FLAG resin | Sigma Aldrich | A4596 | |
| Chemical compound, drug | Resiniferatoxin | Sigma Aldrich | R8756 | |
| Chemical compound, drug | Bio-Beads SM-2 | BioRad | 152–8920 | |
| Chemical compound, drug | 1,2-dimyristoyl-*sn*-glycero-3-phosphocholine | Avanti Polar Lipids | 850345P | |

*Continued on next page*

*Continued*

| Reagent type (species) or resource | Designation | Source or reference | Identifiers | Additional information |
|---|---|---|---|---|
| Chemical compound, drug | 1-palmitoyl-2-oleoyl-*sn*-glycero-3-phosphocholine (POPC) | Avanti Polar Lipids | 850457C | |
| Chemical compound, drug | 1-palmitoyl-2-oleoyl-*sn*-glycero-3-phosphoethanolamine (POPE) | Avanti Polar Lipids | 850757C | |
| Chemical compound, drug | 1-palmitoyl-2-oleoyl-*sn*-glycero-3-phospho-(1'-*rac*-glycerol) (POPG) | Avanti Polar Lipids | 840457C | |
| Other | Whatman No. one filter paper | Sigma Aldrich | WHA1001325 | |
| Other | UltrAuFoil R1.2/1.3 300-mesh grid | Electron Microscopy Sciences | Q350AR13A | |
| Software, algorithm | MotionCor2 | *Zheng et al. (2017)* | http://msg.ucsf.edu/em/software/motioncor2.html | RRID:SCR_016499 |
| Software, algorithm | GCTF | *Zhang (2016)* | https://www.mrc-lmb.cam.ac.uk/kzhang/ | RRID:SCR_016500 |
| Software, algorithm | RELION 3.0 | *Zivanov et al. (2018)* | https://www2.mrc-lmb.cam.ac.uk/relion/ | RRID:SCR_016274 |
| Software, algorithm | Coot | *Emsley and Cowtan (2004)* | https://www2.mrc-lmb.cam.ac.uk/personal/pemsley/coot/ | RRID:SCR_014222 |
| Software, algorithm | Phenix | *Adams et al. (2010)* | http://phenix-online.org/ | RRID:SCR_014224 |
| Software, algorithm | Molprobity | *Chen et al. (2010)* | http://molprobity.biochem.duke.edu/index.php | RRID:SCR_014226 |
| Software, algorithm | UCSF Chimera | *Pettersen et al. (2004)* | https://www.cgl.ucsf.edu/chimera/ | RRID:SCR_004097 |
| Software, algorithm | Pymol | Shrödinger LLC | https://pymol.org/2/ | RRID:SCR_000305 |
| Other | Cryo-electron microscopy structure of rabbit TRPV2 ion channel | *Zubcevic et al. (2018b)* | PDB ID 5AN8 | PMID:26779611 |
| Other | Cryo-electron microscopy structure of rabbit TRPV2 ion channel | *Zubcevic et al. (2018a)* | EMDB ID EMD-6455 | PMID:26779611 |
| Other | Crystal structure of the TRPV2 ion channel in complex with RTx | *Zubcevic et al. (2018a)* | PDB ID 6BWJ | PMID:29728656 |

## Protein expression and purification

The construct for the RTx sensitive, full-length rabbit TRPV2 (TRPV2$_{RTx}$) was prepared by introducing four point mutations (F470S, L505M, L508T and Q528E) into the synthesized full-length rabbit TRPV2 gene (*Zhang et al., 2016*). The construct was cloned into a pFastBac vector with a C-terminal FLAG affinity tag and used for baculovirus production according to manufacturers' protocol (Invitrogen, Bac-to-Bac). The protein was expressed by infecting Sf9 cells with baculovirus at a density of 1.3 M cells ml$^{-1}$ and incubating at 27°C for 72 hr in an orbital shaker. Cell pellets were collected after 72 hr and resuspended in buffer A (50 mM TRIS pH8, 150 mM NaCl, 2 mM CaCl$_2$, 1 μg ml$^{-1}$ leupeptin, 1.5 μg ml$^{-1}$ pepstatin, 0.84 μg ml$^{-1}$ aprotinin, 0.3 mM PMSF, 14.3 mM β-mercaptoethanol, and DNAse I) and broken by sonication (3 × 30 pulses).

For the amphipol-reconstituted TRPV2 (TRPV2$_{RTx-APOL}$) sample, the lysate was supplemented with 40 mM Dodecyl β-maltoside (DDM, Anatrace), 4 mM Cholesteryl Hemisuccinate (CHS, Anatrace) and 2 μM RTx and incubated at 4°C for 1 hr. Insoluble material was removed by centrifugation (8000 g, 30 min), and anti-FLAG resin was added to the supernatant for 1 hr at 4°C.

After binding, the anti-FLAG resin was loaded onto a Bio-Rad column and a wash was performed with 10 column volumes of buffer B (50 mM TRIS pH8, 150 mM NaCl, 2 mM CaCl$_2$, 1 mM DDM, 0.1 mM CHS, 0.1 mg ml$^{-1}$ 1,2-dimyristoyl-*sn*-glycero-3-phosphocholine (DMPC, Avanti Polar Lipids), 2 µM RTx) before elution in five column volumes of buffer C (50 mM TRIS pH8, 150 mM NaCl, 2 mM CaCl$_2$, 1 mM DDM, 0.1 mM CHS, 0.1 mg ml$^{-1}$ DMPC, 2 µM RTx, 0.1 mg ml$^{-1}$ FLAG peptide).

The eluate was concentrated and further purified by gelfiltration on a Superose six column. The peak fractions were collected, mixed with Amphipol A8-35 (Anatrace) in a 1:10 ratio and incubated for 4 hr at 4°C. Subsequently, Bio-Beads SM-2 (Bio-Rad) were added to a 50 mg ml$^{-1}$ concentration and incubated at 4°C overnight to remove detergent.

After reconstitution, the protein was subjected to a second round of gelfiltration on a Superose six column in buffer D (50 mM TRIS pH8, 150 mM NaCl, 2 µM RTx), the peak fractions were collected and concentrated to 2–2.5 mg ml$^{-1}$ for cryo-EM.

For the nanodisc reconstituted TRPV2 (TRPV2$_{RTx-ND}$), the lysate was supplemented with 40 mM Dodecyl β-maltoside (DDM, Anatrace) and 2 µM RTx and incubated at 4° C for 1 hr. The solution was cleared by centrifugation (8000 g, 30 min), and anti-FLAG resin was added to the supernatant for 1 hr at 4°C.

After binding, the anti-FLAG resin was loaded onto a Bio-Rad column and a wash was performed with 10 column volumes of buffer B$_{noCHS}$ (50 mM TRIS pH8, 150 mM NaCl, 2 mM CaCl$_2$, 1 mM DDM, 0.1 mg ml$^{-1}$ DMPC, 2 µM RTx) before elution in five column volumes of buffer C$_{noCHS}$ (50 mM TRIS pH8, 150 mM NaCl, 2 mM CaCl$_2$, 1 mM DDM, 0.1 mg ml$^{-1}$ DMPC, 2 µM RTx, 0.1 mg ml$^{-1}$ FLAG peptide).

The eluate from the anti-FLAG resin was concentrated to ~500 µl. A 10 mg ml$^{-1}$ 3:1:1 mixture of lipids 1-palmitoyl-2-oleoyl-*sn*-glycero-3-phosphocholine (POPC), 1-palmitoyl-2-oleoyl-*sn*-glycero-3-phosphoethanolamine (POPE), 1-palmitoyl-2-oleoyl-*sn*-glycero-3-phospho-(1'-*rac*-glycerol) (POPG) was dried under argon, resuspended in 1 ml 50 mM Tris pH8, 150 mM NaCl and clarified by extrusion, before being incubated for 1 hr with 10 mM DDM. The membrane scaffold protein MSP2N2 was prepared as previously described (*Ritchie et al., 2009*). The concentrated TRPV2 was combined with MSP2N2 and the prepared lipid mixture in a 1:3:200 ratio and incubated at 4°C for 1 hr. After the initial incubation, 50 mg ml$^{-1}$ Bio-Beads SM-2 were added and the mixture was incubated for another hour at 4°C, following which the reconstitution mixture was transferred to a fresh batch of Bio-Beads SM-2 at 50 mg ml$^{-1}$ and incubated overnight at 4°C. Finally, the reconstituted channels were subjected to gelfiltration on Superose six in buffer D, the peak fractions collected and concentrated to 2–2.5 mg ml$^{-1}$ for cryo-EM.

## Cryo-EM sample preparation

TRPV2$_{RTx-APOL}$ and TRPV2$_{RTx-ND}$ were frozen using the same protocol. Before freezing, the concentrated protein sample was supplemented with 300 µM RTx and incubated ~30 min at 4°C. 3 µl sample was dispensed on a freshly glow discharged (30 s) UltrAuFoil R1.2/1.3 300-mesh grid (Electron Microscopy Sciences), blotted for 3 s with Whatman No. one filter paper using the Leica EM GP2 Automatic Plunge Freezer at 23°C and >85% humidity and plunge-frozen in liquid ethane cooled by liquid nitrogen.

## Cryo-EM data collection

Data for both TRPV2$_{RTx-APOL}$ and TRPV2$_{RTx-ND}$ were collected using the Titan Krios transmission electron microscope (TEM) operating at 300 keV using a Falcon III Direct Electron Detector operating in counting mode at a nominal magnification of 75,000x corresponding to a physical pixel size of 1.08 Å/pixel.

For the TRPV2$_{RTx-APOL}$ 1293 movies (30 frames/movie) were collected using a 60 s exposure with an exposure rate of ~0.8 e$^-$/pixel/s, resulting in a total exposure of 42 e$^-$/Å (*Cao et al., 2013a*) and a nominal defocus range from −1.25 µm to −3.0 µm.

For TRPV2$_{RTx-ND}$, 2254 movies were collected (30 frames/movie) with 60 s exposure and exposure rate of ~0.8 e$^-$/pixel/s. The total exposure was of 42 e$^-$/Å (*Cao et al., 2013a*) and a nominal defocus range from −1.25 µm to −3.0 µm.

## Reconstruction and refinement

*TRPV2$_{RTx-APOL}$* MotionCor2 (*Zheng et al., 2017*) was used to perform motion correction and dose-weighting on 1293 movies. Unweighted summed images were used for CTF determination using GCTF (*Zhang, 2016*). Following motion correction and dose-weighting and CTF determination, micrographs which contained Figure of Merit (FoM) values of <0.12 and astigmatism values > 400 were removed, leaving 1207 micrographs for further analysis. An initial set of 1660 particles was picked manually and subjected to reference-free 2D classification (k = 12, T = 2) which was used as a template for automatic particle picking from the entire dataset (1207 micrographs). This yielded a stack of 580,746 particles that were binned 4 × 4 (4.64 Å/pixel, 64 pixel box size) and subjected to reference-free 2-D classification (k = 58, T = 2) in RELION 3.0 (*Zivanov et al., 2018*). Classes displaying the most well-defined secondary structure features were selected (470,760 particles) and an initial model was generated from the 2D particles using the Stochastic Gradient Descent (SGD) algorithm as implemented in RELION 3.0. 3D auto-refinement in RELION 3.0 was performed on the 470,760 particles with no symmetry imposed (C1), using the initial model, low-pass filtered to 30 Å, as a reference map. This resulted in an 8.9 Å 3D reconstruction, which was then used for re-extraction and re-centering of 2 × 2 binned particles (2.16 Å/pixel, 128 pixel box size). 3D classification (k = 4, T = 8) without imposed symmetry (C1) was performed on the extracted particles, using a soft mask calculated from the full molecule. Classes 2–4 (90,862, 109,623 and 101,570 particles, respectively) all possessed well-defined secondary structure, but visual inspection of the maps suggested that the classes represented distinct conformational states. Therefore, each class was processed separately. For each class, the particles were extracted and unbinned (1.08 Å/pixel, 256 pixel box size), and soft masks calculated. 3D auto-refinement of the individual classes without symmetry imposed (C1) yielded 4.7 Å (class 2), 3.6 Å (class 3) and 3.2 Å (class 4) 3D reconstructions. Inspection of these volumes revealed that classes 2 and 3 adopted two-fold (C2) symmetry, while class four was four-fold symmetric (C4). Particles from class 2 were subjected to particle movement and dose-weighting using the 'particle polishing' function as implemented in RELION 3.0. The shiny particles were input into 3D auto-refinement with a soft mask and C2 symmetry applied, resulting in a 4.19 Å reconstruction (TRPV2$_{RTx-APOL\ 3}$). Similarly, particles from class 3 were subjected to polishing, and the following 3D auto-refinement with a soft mask and C2 symmetry applied resulted in a 3.3 Å final reconstruction (TRPV2$_{RTx-APOL2}$). Particles from class 4 were first subjected to CTF refinement using the 'CTF refine' feature in RELION 3.0. Particle polishing was then performed, followed by 3D auto-refinement with a soft mask and C4 symmetry applied, yielding a 2.91 Å reconstruction (TRPV2$_{RTx-APOL\ 1}$). All resolution estimates were based on the gold-standard FSC 0.143 criterion (*Scheres and Chen, 2012*; *Chen et al., 2013*).

*TRPV2$_{RTx-ND}$* The 2254 collected movies were subjected to motion correction and dose-weighting (MotionCor2) and CTF estimation (GCTF) in RELION 3.0. Micrographs with FoM values < 0.13 and astigmatism values > 400 were removed, resulting in a selection of 1580 good micrographs. From these, 2015 particles were picked manually, extracted (without binning, 1.08 Å/pixel, 256 pixel box size) and subjected to reference-free 2D classification (k = 12, T = 2) that was used as a template for autopicking. This resulted in a 1,407,292 stack of particles that were binned 4 × 4 (4.32 Å/pixel, 64 pixel box size) and subjected to reference-free 2D classification (k = 100, T = 2). Classes exhibiting the most well-defined secondary structure features were selected, resulting in 482,602 particles. These were re-extracted (2 × 2 binned, 2.16 Å/pixel, 128 pixel box size) and put into 3D auto-refinement, using the previously obtained map of apo TRPV2 (EMD-6455) filtered to 30 Å as a reference with no symmetry applied (C1). The 3D auto-refinement yielded a 5.4 Å reconstruction. The particles were then subjected to 3D classification (k = 6, T = 8), with a soft mask and the 5.4 Å volume as a reference without imposed symmetry (C1). Only two of the six classes (classes 1 and 6) contained significant density in the TM domains. They were selected (112,622 particles), re-extracted, re-centered and unbinned (1.08 Å/pixel, 256 pixel box size) before being input into 3D auto-refinement without symmetry imposed (C1) and with a soft mask and the previous 5.4 Å reconstruction filtered to 30 Å as a reference. The 3D auto-refinement resulted in a 4.12 Å map, which was then subjected to Bayesian particle polishing. 3D auto-refinement was then performed on the resulting shiny particles with no symmetry applied (C1), resulting in a 4 Å reconstruction. The particles were then subjected to CTF refinement, yielding a 3D reconstruction resolved to 4 Å (C1). However, visual inspection of the map revealed a strong tendency towards two-fold symmetry. Therefore, 3D auto-

refinement was repeated with C2 symmetry applied, resulting in a map resolved to 3.84 Å as estimated by gold-standard FSC 0.143 criterion.

## Model building

The TRPV2$_{RTx-APOL}$ and TRPV2$_{RTx-ND}$ models were built into the cryo-EM electron density map in Coot (*Emsley and Cowtan, 2004*), using the structures of TRPV2 (PDB 5AN8 and 6BWJ) as templates. The structures were real-space refined in Coot, and iteratively refined using the phenix.real_space_refine as implemented in the Phenix suite (*Adams et al., 2010*). Structures were refined using global minimization and rigid body, with high weight on ideal geometry and secondary structure restraints. The Molprobity server (*Chen et al., 2010*) (http://molprobity.biochem.duke.edu/) was used to identify problematic areas, which were subsequently manually rebuilt. The radius of the permeation pathways was calculated using HOLE (*Smart et al., 1996*). All analysis and structure illustrations were performed using Pymol (The PyMOL Molecular Graphics System, Version 2.0) and UCSF Chimera (*Pettersen et al., 2004*).

## Acknowledgements

Cryo-EM data were collected at the Shared Materials Instrumentation Facility at Duke University as part of the Molecular Microscopy Consortium, and screening was performed at National Institute of Environmental Health Sciences (NIEHS). We thank Alberto Bartesaghi for developing a pre-processing interface, which enabled on-the-fly monitoring of cryo-EM image quality during data collection. Funding: This work was supported by the National Institutes of Health (R35NS097241 to S-YL) and by the National Institutes of Health Intramural Research Program; US National Institute of Environmental Health Sciences (ZIC ES103326 to MJB). The EM maps and atomic models have been deposited with the Electron Microscopy Data Bank (accession numbers EMD-20143, EMD-20145, EMD-20146, and EMD-20148) and the Protein Data Back (entry codes 6OO3, 6OO4, 6OO5, and 6OO7), respectively.

## Additional information

### Funding

| Funder | Grant reference number | Author |
|--------|------------------------|--------|
| National Institute of Neurological Disorders and Stroke | R35NS097241 | Seok-Yong Lee |
| National Institute of Environmental Health Sciences | ZIC ES103326 | Mario J Borgnia |

The funders had no role in study design, data collection and interpretation, or the decision to submit the work for publication.

### Author contributions

Lejla Zubcevic, Conceptualization, Data curation, Validation, Writing—original draft, Writing—review and editing; Allen L Hsu, Data curation, Validation, Writing—review and editing; Mario J Borgnia, Supervision, Funding acquisition, Validation, Writing—review and editing; Seok-Yong Lee, Conceptualization, Supervision, Funding acquisition, Validation, Writing—original draft, Writing—review and editing

### Author ORCIDs

Lejla Zubcevic (iD) https://orcid.org/0000-0002-1884-9235
Allen L Hsu (iD) http://orcid.org/0000-0003-2065-3802
Mario J Borgnia (iD) https://orcid.org/0000-0001-9159-1413
Seok-Yong Lee (iD) https://orcid.org/0000-0002-0662-9921

### Decision letter and Author response

Decision letter https://doi.org/10.7554/eLife.45779.046

Author response https://doi.org/10.7554/eLife.45779.047

## Additional files

### Supplementary files
• Transparent reporting form
DOI: https://doi.org/10.7554/eLife.45779.028

### Data availability
The EM maps and atomic models have been deposited with the Electron Microscopy Data Bank (accession numbers EMD-20143, EMD-20145, EMD-20146, and EMD-20148) and the Protein Data Back (entry codes 6OO3, 6OO4, 6OO5, and 6OO7), respectively.

The following datasets were generated:

| Author(s) | Year | Dataset title | Dataset URL | Database and Identifier |
|---|---|---|---|---|
| Zubcevic L, Hsu AL, Borgnia MJ, Lee S-Y | 2019 | Cryo-EM structure of the C4-symmetric TRPV2/RTx complex in amphipol resolved to 2.9 A | http://www.ebi.ac.uk/pdbe/entry/emdb/EMD-20143 | Electron Microscopy Data Bank, EMD-20143 |
| Zubcevic L, Hsu AL, Borgnia MJ, Lee S-Y | 2019 | Cryo-EM structure of the C2-symmetric TRPV2/RTx complex in amphipol resolved to 3.3 A | http://www.ebi.ac.uk/pdbe/entry/emdb/EMD-20145 | Electron Microscopy Data Bank, EMD-20145 |
| Zubcevic L, Hsu AL, Borgnia MJ, Lee S-Y | 2019 | Cryo-EM structure of the C2-symmetric TRPV2/RTx complex in amphipol resolved to 4.2 A | http://www.ebi.ac.uk/pdbe/entry/emdb/EMD-20146 | Electron Microscopy Data Bank, EMD-20146 |
| Zubcevic L, Hsu AL, Borgnia MJ, Lee S-Y | 2019 | Cryo-EM structure of the C2-symmetric TRPV2/RTx complex in nanodiscs | http://www.ebi.ac.uk/pdbe/entry/emdb/EMD-20148 | Electron Microscopy Data Bank, EMD-20148 |
| Zubcevic L, Hsu AL, Borgnia MJ, Lee S-Y | 2019 | Cryo-EM structure of the C4-symmetric TRPV2/RTx complex in amphipol resolved to 2.9 A | http://www.rcsb.org/structure/6OO3 | Protein Data Bank, 6OO3 |
| Zubcevic L, Hsu AL, Borgnia MJ, Lee S-Y | 2019 | Cryo-EM structure of the C2-symmetric TRPV2/RTx complex in amphipol resolved to 3.3 A | http://www.rcsb.org/structure/6OO4 | Protein Data Bank, 6OO4 |
| Zubcevic L, Hsu AL, Borgnia MJ, Lee S-Y | 2019 | Cryo-EM structure of the C2-symmetric TRPV2/RTx complex in amphipol resolved to 4.2 A | http://www.rcsb.org/structure/6OO5 | Protein Data Bank, 6OO5 |
| Zubcevic L, Hsu AL, Borgnia MJ, Lee S-Y | 2019 | Cryo-EM structure of the C2-symmetric TRPV2/RTx complex in nanodiscs | http://www.rcsb.org/structure/6OO7 | Protein Data Bank, 6OO7 |

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
