## [Decision Letter]

Thank you for submitting your article "Symmetry transitions during gating of the TRPV2 ion channel in lipid membranes" for consideration by *eLife*. Your article has been reviewed by three peer reviewers, and the evaluation has been overseen by Kenton Swartz as the Reviewing Editor and Richard Aldrich as the Senior Editor. The following individuals involved in review of your submission have agreed to reveal their identity: Richard K Hite (Reviewer #3); Raimund Dutzler (Reviewer #4).

The reviewers have discussed the reviews with one another and the Reviewing Editor has drafted this decision to help you prepare a revised submission.

Summary:

In this report by Zubcevic and colleagues, single-particle cryo-EM structures of TRPV2 in amphipols and lipid nanodiscs in complex with the agonist resiniferatoxin (RTx) are presented. Three-dimensional classification of the amphipol-reconstituted channels reveal three distinct conformations, two of which display C2 symmetry and a third which displays C4 symmetry. Classification of the nanodisc-reconstituted channels reveal just a single conformation which displays C2 symmetry. Previously, the same group had reported a ligand-free, closed cryo-EM structure of TRPV2 which displayed C4 symmetry and a RTx-bound C2 symmetric crystal structure with an open selectivity filter gate that displayed two-fold symmetry and a closed common gate that displayed four-fold symmetry. Comparison of the ampiphol-reconstituted structures with the previous structures indicated that in all three conformations the TM domain adopts a nearly four-fold symmetric structure that resembles the ligand-free, closed state, while the ARD is flexible and adopts a range of different conformations. In contrast, large conformational changes are seen throughout the nanodisc-reconstituted structure. The selectivity filter gate is opened in a 2-fold symmetric fashion by alternating rotations of the pore helices in adjacent subunits. The common gate also adopts a 2-fold symmetric conformation, with two of the subunits adopting a closed state and two adopting an open state. Comparison of the nanodisc-reconstituted sample with the previous RTx-bound crystal structure suggested that they represent distinct conformational states with the crystal structure which contained a fully closed common gate potentially representing an earlier state in the transition from closed to an open state. This work provides evidence that RTx binding to TRPV2 induces two-fold symmetric pre-activated states in a lipid bilayer environment and will be of general interest. We request that you address the following issues in revision.

Essential revisions:

1) Some of the conformational differences observed between the structures or between subunits from a single structure are small (see Figure 3—figure supplement 1C, D; Figure 3—figure supplement 2B, C), or reflect changes in the patterns of hydrogen bonding such as the α- to π-helix transitions. Given that the resolution of the structures is not very high, the authors should provide structural comparisons including the experimental density maps to support their proposed conformational differences, rather than simply using the structural models. The density maps should be used to provide stronger evidence that the pattern of main-chain hydrogen bonds has indeed been disrupted in the S4-S5 and S6 helices to give rise to π-helical bulges. As symmetry is a major feature of this work, it would helpful to compare the model refined using the symmetrized reconstructions in density maps calculated without symmetry. This is particularly important for the selectivity filter and common gates of the four subunits in the nanodisc-reconstituted reconstruction.

2) Why is map-to-model fit so poor for TRPV2_RTx-APOL1_? The map-to-model fit has a 0.5 correlation at approximately 4Å while the FSC between half-maps is approximately 2.9 Å. How was this model refined into the density map? What does nominal resolution mean? Why is it different from FSC=0.143 in Table 1 and why are the unmasked resolution estimates higher than the masked resolution estimates? Any masks used for classification or refinement should be displayed in the figure supplements.

3) The authors suggest that the ampiphol restrict the flexibility of the TM domain. It would be helpful to compare the belt of ampiphol-reconstituted samples with those of the nanodisc-reconstituted samples to see if sufficient space is available for the channel to adopt the conformation of the nanodisc-reconstituted channels. Were lipid molecules resolved in the nanodisc structures that may provide the flexibility required to adopt the different conformation?

4) It would be interesting to know how many particles contribute to the different 3D classes displayed Figure 1—figure supplement 2. Do the particles in class 1 shown in Figure 1—figure supplement 1 really display and unsymmetrical (C1) channel and can we learn anything from this low-resolution structure. While it is possible that the X-ray structure of TRPV2-RTx complex could be an intermediate on the activation pathway, the authors might also consider the possibility that the pronounced C2-symmetric features observed in this structure would reflect the conditions in the crystalline environment and might thus not be an intermediate on the activation path.

5) Even though RTx activates both TRPV1 and TRPV2 with an extremely high affinity and efficacy, the kinetics of activation are distinctly slow, even when compared to other vanilloids with lower affinity such as capsaicin in the case of TRPV1. This raises the possibility that RTx can interact with the pocket in multiple orientations, with only some of them leading to channel activation. At present, it is not possible to determine whether the structures observed in this study are indeed reflecting RTx-channel complexes that are components of the activation pathway, or whether they reflect off-pathway intermediates that are not conducing to activation. It is intriguing that no open state has been determined even though RTx activates TRPV2-QM with very high affinity and efficacy. Are the binding poses of RTx inferred from the densities different in all structures? Are they different between subunits in a given structure?

6) The text makes constant reference to a selectivity filter activation gate in these channels. However, there is currently no functional evidence indicating that the selectivity filter does indeed function as an activation gate. This should be reflected in the text.

7) In several points throughout the manuscript (subsection “RTx induces a break in symmetry in TRPV2_RTx-ND_”, last paragraph), the authors mention that their previous work demonstrated that a network of hydrogen bonds between the pore helix, the S5 and the S6 is essential for activation. However, the results in the previous publication do not provide conclusive evidence for the involvement of hydrogen bonds, but instead show that alanine mutations in that region affect cation permeation through an unknown mechanism that could involve a network of hydrogen bonds. This should be clearly stated in the manuscript.

8) In the third paragraph of the Discussion, the authors mention that their previous work has shown that C2 symmetric states are critical for permeation of large organic cations. We think it is more accurate to state that their data suggests that C2 symmetry could be involved in channel activation by RTx, but it does not provide conclusive evidence that C2 symmetries are directly connected to channel activation or to the mechanism of cation permeation in an open channel. No experiments were done in that study to directly probe the functional consequences of the proposed asymmetries in the tetrameric channel.

---

## [Author Response]

Essential revisions:1) Some of the conformational differences observed between the structures or between subunits from a single structure are small (see Figure 3—figure supplement 1C, D; Figure 3—figure supplement 2B, C), or reflect changes in the patterns of hydrogen bonding such as the α- to π-helix transitions. Given that the resolution of the structures is not very high, the authors should provide structural comparisons including the experimental density maps to support their proposed conformational differences, rather than simply using the structural models. The density maps should be used to provide stronger evidence that the pattern of main-chain hydrogen bonds has indeed been disrupted in the S4-S5 and S6 helices to give rise to π-helical bulges.

We have now included figure supplements showing the density around the π-helices in all structures (Figure 3—figure supplement 4). However, we agree with the reviewer that the resolution of the TRPV2_RTx-ND_ of the map is not very high enough to define π-helix on S6 unambiguously and thus we have removed references in the text and figures to the S6 π-helix in structure throughout the text, and refer them to as bent helices instead.

As symmetry is a major feature of this work, it would helpful to compare the model refined using the symmetrized reconstructions in density maps calculated without symmetry. This is particularly important for the selectivity filter and common gates of the four subunits in the nanodisc-reconstituted reconstruction.

We thank reviewers for the great suggestion. During processing, we merged classes 1 and 6, which displayed a high overall similarity (CC=0.98) and then proceeded with refinement with C1 imposed. Because visual inspection indicated that this reconstruction possesses two-fold symmetry, we further refined the combined particles with C2 symmetry imposed. We have now also calculated the symmetry of the map combining classes 1 and 6 using the “Map Symmetry” function implemented in the Phenix suite. The best solution is C2 symmetry (score 1.28, CC=0.90). (Score reflects the square root of the number of elements in the symmetry multiplied by the map correlation for that symmetry, so the max score for C2 symmetry is 1.414 and 2 is the max score for C4 symmetry). We also calculated the score for C4 symmetry (score=1.6, CC=0.80) which indicates that C2 symmetry is more appropriate for this map. We have now also made a remark in the manuscript describing this analysis (subsection “RTx induces a break in symmetry in TRPV2_RTx-ND_”, first paragraph).

For the purpose of this review, we have now included an FSC curve which estimates the fit of the model, originally built into the C2 map, into the C1 reconstruction. In addition, we have now also performed separate refinements of classes 1 and 6 with no symmetry imposed and estimated the fit of the TRPV2_RTx-ND_ atomic model, built into the C2 symmetry, to these non-symmetrized maps. This data is shown in Figure 1—figure supplement 3. Figure 1—figure supplement 3 now also shows the density around the critical regions (pore helices and S6 helices) in the class 1+class 6 non-symmetrized map. A mention of this is now included in the manuscript (see the second paragraph of the aforementioned subsection).

2) Why is map-to-model fit so poor for TRPV2_RTx-APOL1_? The map-to-model fit has a 0.5 correlation at approximately 4Å while the FSC between half-maps is approximately 2.9 Å. How was this model refined into the density map? What does nominal resolution mean? Why is it different from FSC=0.143 in Table 1 and why are the unmasked resolution estimates higher than the masked resolution estimates? Any masks used for classification or refinement should be displayed in the figure supplements.

We thank the reviewers for pointing this out. A mistake was made in the plotting of the APOL FSC curves during our initial manuscript preparation. This has now been corrected and the new FSC curves placed in Figure 1—figure supplement 1.

3) The authors suggest that the ampiphol restrict the flexibility of the TM domain. It would be helpful to compare the belt of ampiphol-reconstituted samples with those of the nanodisc-reconstituted samples to see if sufficient space is available for the channel to adopt the conformation of the nanodisc-reconstituted channels. Were lipid molecules resolved in the nanodisc structures that may provide the flexibility required to adopt the different conformation?

We have now included a figure (Figure 4—figure supplement 6) showing the amphipol and nanodisc clouds surrounding the channels in the two different reconstitution conditions. We used the common nanodisc scaffold protein MSP2N2, which has been successfully used in many structural studies of TRP channels. MSP2N2 has an estimated diameter of ~17 nm and is large enough to accommodate TRPV2. The size of nanodisc is largely defined by the scaffold protein while amphipol polymers surround membrane proteins in a way that is similar to detergents. The quality of EM density for the nanodisc cloud is low and neither the scaffold protein nor the bound lipid molecules are well resolved. It is therefore difficult to say whether the lipids stabilize the captured conformation. However, a side-by-side comparison of the amphipol and nanodisc structures indicates that the nanodisc adopts a distinctly oval shape to accommodate the TRPV2-RTx complex, while amphipol retains a squarer form.

4) It would be interesting to know how many particles contribute to the different 3D classes displayed Figure 1—figure supplement 2.

This has now been included in the figure.

Do the particles in class 1 shown in Figure 1—figure supplement 1 really display and unsymmetrical (C1) channel and can we learn anything from this low-resolution structure.

We did indeed try to refine this class, but the resolution never got past ~7 Å. We therefore did not feel comfortable presenting this data or commenting on its symmetry and conformational state. We have now included a sentence stating this in the manuscript.

While it is possible that the X-ray structure of TRPV2-RTx complex could be an intermediate on the activation pathway, the authors might also consider the possibility that the pronounced C2-symmetric features observed in this structure would reflect the conditions in the crystalline environment and might thus not be an intermediate on the activation path.

This point was addressed in the paper published in NSMB (Zubcevic et al., 2018). In brief, we conducted electrophysiological experiments to probe the validity of the structure and found that the interactions that were *uniquely* found in the TRPV2-RTx crystal structure were of critical importance for channel function and that the structure was therefore not an artifact of crystallization. We also discussed the possibility of crystallization artifacts in the Introduction of the current manuscript.

5) Even though RTx activates both TRPV1 and TRPV2 with an extremely high affinity and efficacy, the kinetics of activation are distinctly slow, even when compared to other vanilloids with lower affinity such as capsaicin in the case of TRPV1. This raises the possibility that RTx can interact with the pocket in multiple orientations, with only some of them leading to channel activation. At present, it is not possible to determine whether the structures observed in this study are indeed reflecting RTx-channel complexes that are components of the activation pathway, or whether they reflect off-pathway intermediates that are not conducing to activation. It is intriguing that no open state has been determined even though RTx activates TRPV2-QM with very high affinity and efficacy. Are the binding poses of RTx inferred from the densities different in all structures? Are they different between subunits in a given structure?

We have observed that RTx can bind to TRPV2 in different modes, which we believe contributes significantly to the observed C2 symmetry. We have addressed this point in our previous publication as well as this study, but our clarification may have been somewhat insufficient. We have now included a figure supplement to show the binding poses of RTx in the different structures as well as in the subunits of different structures. In brief, in the C4 TRPV2_RTx-APOL 1_ structure, the RTx binding poses are the same in all subunits. They are also nearly identical in TRPV2_RTx-APOL 2_. However, they differ from the ones captured in the C4 structure. In both TRPV2_RTx-APOL 2_ andTRPV2_RTx-ND,_the binding poses are different within the subunits and they also differ from the poses observed in both TRPV2_RTx-APOL 1_ and TRPV2_RTx-APOL 2_. Nevertheless, given the resolution of TRPV2_RTx-APOL 3_ and TRPV2_RTx-ND_, the position of RTx in these structures should be taken with caution.

The point concerning the slow activation kinetics of RTx is excellent. In order to allow ample time between adding RTx to freezing the grids, we included RTx in our buffers from the very beginning of the purification. This is in contrast to the Julius-Cheng TRPV1/RTx/DkTx complex where the toxin was added to the sample ~30 minutes before freezing.

The point questioning whether the RTx-channel complexes observed here are components of the activation pathway is equally valid. The Swartz lab study that originally transplanted RTx-sensitivity into TRPV2 (Zhang et al., 2016) conducted recordings of single RTx-activated TRPV2 channels and found that these can achieve a high (~0.9) open probability. We hypothesize that the absence of a fully open state may point to the absence of unknown factors in our experimental conditions (e.g. the presence of the membrane bilayer rather than nanodisc). This is in line with observations made by many groups that agonist-bound ion channel structures often do not adopt fully open states.

However, we believe that the absence of the open state does not dismiss the rest of our findings as “off-pathway” conformations – the conformational changes that we observe in both the amphipol and nanodisc structures are in line with conformations associated with activation of closely related channels: the ARD rotation seen in TRPV1 (Cao et al., 2013; Liao et al., 2013) and the α-to-π transition in the S4-S5 linker in TRPV2 (Zubcevic et al., 2018). Notably, voltage-gated cation channels can assume multiple closed states before opening, and it is possible that TRPV2 channels also traverse a number of intermediate non-conducting states before entering the open conformation. It’s also worth pointing out that all of our data on the TRPV2/RTx complex, both crystallographic and cryo-EM, contains the same hot spots for conformational change: the S4-S5 linker and the pore helix. By contrast, pore of related channels TRPV3 and TRPV6 do not exhibit any change in the conformation of the pore helix during gating. And in addition, the pore of TRPV6 is preserved even in the structure of the point mutant which induces a non-domain swapped conformation in the channel, indicating that in TRPV6 the pore is a stable unit- as expected for a highly selective channel. In light of this, our TRPV2/RTx data points to the inherent flexibility of the pore helix in this channel.

Furthermore, we believe it unlikely that two-fold symmetry is an experimental artifact, as it’s only been observed in a few select TRP channels. Crystal and cryo-EM structures of K^+^ channels have never observed two-fold symmetry in the channel pores, and even the recent cryo-EM study of the K_V_ channel in nanodiscs (Matthies et al., 2018) is identical to the four-fold symmetric crystal structure solved by the MacKinnon lab. Along the same lines, the structures of Ca^2+^-selective TRPV5 and TRPV6, which have been solved in many different conditions (e.g. crystallization, cryo-EM, agonist and antagonist-bound, detergent, nanodisc, amphipol, mutations) have never been captured in non-C4 symmetric arrangements. These, like K^+^ channels, possess seemingly rigid pore structures (meaning extensive interactions between pore helices and the rest of pore), which is consistent with their observed strong preference for four-fold symmetry.

We think that the ability of TRPV2 to enter different two- and four-fold symmetric states is a reflection of the intrinsic flexibility in the TRPV2 channel pore and especially the pore helix. As we discussed in our previous paper, we think that this flexibility might be unique to TRPV1 and TRPV2 amongst TRPV channels and that it may endow them with the ability to permeate large organic cations.

At its core, the question posed here applies universally to all structural data: how well do cryo-EM or crystallization conditions sample the conformational landscape of proteins in their native environments? In this manuscript, we have evaluated our cryo-EM structures based our structural and functional data as well as the available literature and proposed a hypothetical sequence of events that may lead to TRPV2 channel opening.

6) The text makes constant reference to a selectivity filter activation gate in these channels. However, there is currently no functional evidence indicating that the selectivity filter does indeed function as an activation gate. This should be reflected in the text.

That is an excellent point for discussion. To make our long answer short, we think that the SF acts as an activation gate in TRPV1 and TRPV2, but not in other TRPV members (TRPV3-TRPV6). The references to the “selectivity filter gate” originated with the very first TRPV1 structure which showed that the channel possesses two restrictions in the non-conductive state: the methionine in the SF which seals the selectivity filter (dubbed the “upper gate”) and the isoleucine in S6 which creates a hydrophobic seal at the common gate (dubbed the “lower gate”). In the RTx/DkTx-bound TRPV1, both of these restrictions are removed: the common gate opens via a bend in the S6 and the SF via a tilt of the pore helix. While it is possible that the SF opening is due to DkTx binding or simply reflects a conformational change that is coupled to the opening of the “lower gate”, the structural data points to the existence of the SF gate. We agree with the reviewer that the functional evidence for the SF activation gate is lacking. From a structural standpoint, we reason that TRPV2, like TRPV1, possesses a SF activation gate: the architecture of TRPV2 is very similar to TRPV1 and shows two restrictions in the closed state and binding of RTx induces an open conformation of the SF gate (that the pore helices tilt significantly and move away from the ion permeation pathway).

However, we think that other TRPV channels (TRPV3-6) only possess one restriction in their pores: the common gate. Structural studies of these channels have not observed significant movement around the PH or the SF during gating (e.g. TRPV3), which is consistent with the absence of a SF gate. We have now included a paragraph (subsection “RTx induces a break in symmetry in TRPV2_RTx-ND_”, last paragraph) to address this point.

7) In several points throughout the manuscript (subsection “RTx induces a break in symmetry in TRPV2_RTx-ND_”, last paragraph), the authors mention that their previous work demonstrated that a network of hydrogen bonds between the pore helix, the S5 and the S6 is essential for activation. However, the results in the previous publication do not provide conclusive evidence for the involvement of hydrogen bonds, but instead show that alanine mutations in that region affect cation permeation through an unknown mechanism that could involve a network of hydrogen bonds. This should be clearly stated in the manuscript.

We believe that our reference to the hydrogen bond network between the threonine in the pore helix and tyrosine residues in S5 and S6 is chemically sound. The donor/acceptor are within 3Å of each other in our 3.1 Å crystal structure, and therefore classified as a hydrogen bond of moderate strength (like most hydrogen bonds in proteins). While it is true that hydrogens are not resolved at ~3 Å, this assumption is widely used in structural biology (i.e. the hydrogen bonding in α-helices).

No other type of major interaction can exist between threonine and tyrosine side chains that are within 3Å beside hydrogen bonding. It is true that we performed alanine mutagenesis studies our previous studies (Zubcevic et al., 2018) instead of more subtle mutations such as threonine to valine (isosteric mutation) or tyrosine to phenylalanine (hydroxyl group removal). However, if we apply the same strict standards, introducing above-mentioned mutations to probe H-bonding would also be insufficient evidence for the involvement of H-bonding, and one would at least need to conduct an unnatural amino acid mutagenesis study of TRPV2 to properly probe the H-bonds. In response to the reviewer’s request, throughout the text, we have clarified this point by stating that the interaction between the Thr and Tyr residues (which includes H-bonds), is important for activation.

8) In the third paragraph of the Discussion, the authors mention that their previous work has shown that C2 symmetric states are critical for permeation of large organic cations. We think it is more accurate to state that their data suggests that C2 symmetry could be involved in channel activation by RTx, but it does not provide conclusive evidence that C2 symmetries are directly connected to channel activation or to the mechanism of cation permeation in an open channel. No experiments were done in that study to directly probe the functional consequences of the proposed asymmetries in the tetrameric channel.

This is a great point to bring up, as it allows us to explain our reasoning in more detail. In our NSMB paper we performed a series of experiments to address the role of two-fold symmetry. Our structures showed that RTx-bound TRPV2 adopts a two-fold symmetric state, where the SF takes on two distinct conformations: a narrow conformation, which coordinates Ca^2+^ ions, and a wide conformation, which is wide enough to permeate YO-PRO-1. In this structure, an interaction exits between the pore helix residue Thr602 and S5 helix residue Tyr542 in the subunits with the narrow SF conformation, but it is absent in the subunits in the wide conformation. This interaction is also absent in the ligand-free cryo-EM structure of TRPV2 in C4 symmetry, which adopts a closed conformation of the SF gate. We showed that eliminating the Thr602-Tyr542 interaction, which exists only in the 2-fold symmetric structure, results in channels that have normal Na^+^ currents but are unable to conduct YO-PRO-1. In addition, mutations Thr602A and Tyr542A both decrease the NMDG^+^/Na^+^ permeability ratio (these experiments were done in inside-out patches to avoid ion accumulation artifacts). This was also shown to be true for the corresponding residues in TRPV1. Interestingly, the Thr602-Tyr542 interaction is absent from all four subunits of the RTx/DkTx-bound TRPV1, consistent with fact that the pore of RTX/DkTx-bound TRPV1, due to its narrow profile, is incompatible with permeation of large organic cations.

These experiments showed that:

1) Disruption of the interactions between the SF and the S5 preferentially reduce the permeation of large organic cations;

2) This interaction has only been observed in the channel with C2-symmetry;

3) Intriguingly, this interaction exists in the contracted subunit, not in the widened subunit, further supporting the involvement of reduced symmetry in the ability of the channel to permeate large organic cations.

Therefore, based on our data we believe that reduced symmetry is involved in large organic cation permeation. In our manuscript, we have now changed the text to state that “our previous studies suggest that C2 symmetric states play an important role for permeation of large organic cations”.

Two-fold symmetry has now been observed in a number of TRP channels in a variety of different experimental conditions, but almost always in the presence of agonists: TRPV3 (Zubcevic et al., 2018), TRPV2 (Zubcevic et al., 2018; Pumroy et al., 2019) and TRPM2 (Yin et al., 2019). We believe that further exploration of these experimentally observed symmetry transitions may help us refine our understanding of TRP(V) channel activation and physiology.

References:

Matthies D, Bae C, Toombes GES, Fox T, Bartesaghi A, Subramaniam S, Swartz KJ. *Single-particle cryo-EM structure of a voltage-activated potassium channel in lipid nanodiscs. eLife*. 2018. doi: 10.7554/eLife.37558.

Pumroy RA, Samanta A, Liu Y, Hughes TET, Zhao S, Yudi Y, Huynh KW, Zhou ZH, Rohacs T, Han S, Moiseenkova-Bell VY. *Molecular mechanism of TRPV2 channel modulation by cannabidiol.* bioRxiv, May 24, 2019. Doi: 10.1101/521880

Yin Y, Wu M, Hsu AL, Borschel WF, Borgnia MJ, Lander GC, Lee SY*. Visualizing structural transitions of ligand-dependent gating of the TRPM2 channel.* bioRxiv, Jan 9, 2019. Doi: 10.1101/516468.